# Modular Diffusion Policy Training: Decoupling and Recombining Guidance and Diffusion for Offline RL

## Abstract

In classifier-free guidance(CFG) for offline reinforcement learning(RL), the diffusion model and its guidance are typically trained jointly and applied jointly as the policy network. Before the guidance network has converged, it provides unstable or even misleading gradient shifts for reward optimization. Such strict coupling also prevents the guidance module from being reused across different diffusion models. We propose Guidance-First Diffusion Training (GFDT), which pretrains and freezes the guidance model before diffusion policy learning. This decoupling reduces peak memory and computational overhead by 38.1%, and reduces required diffusion training steps by 65.6% and 27.66% on locomotion and navigation tasks, respectively. Beyond these efficiency gains, the method achieves significant performance improvements of up to 43.16% and 60.98% on these respective offline RL benchmarks. The ablation study on delaying the maturation of the guidance module reduces final diffusion performance by up to 37.7% inverse validates our guidance-first design. Moreover, we uncover a strong plug-and-play property: Cross-algorithm swaps (e.g., Implicit Q-Learning (IDQL) guidance for Diffusion Q-Learning (DQL) policies) perform comparably to the stronger of the two, despite never being co-trained.

Our theoretical analysis also demonstrates that GFDT facilitates convergence to an optimal guidance and accelerates the training process. We further prove that plug-and-play remains valid as long as the guidance and the diffusion model are trained with the same data distribution. Limitations arising from dataset mismatch are analyzed in detail, which further underscores the necessity of distributional alignment. This work opens a new line of research by treating diffusion and guidance as modular units that can be recombined, rather than as a monolithic process, suggesting a paradigm that may guide the future development of diffusion-based RL.

## 1 Introduction

By formulating policy learning as a conditional generative process, **diffusion policies** are capable of modeling complex, multimodal behaviors. Building on this perspective, diffusion-based policies have emerged as a powerful paradigm for behavior generation in offline decision-making. Diffusion policies for offline RL are typically composed of two core modules: a generative diffusion model that produces actions through iterative denoising, and a guidance module that provides gradient bias to steer generation toward higher-reward behavioral modes. This design is known as the **CFG** paradigm(Ho & Salimans, 2022). This guidance–diffusion paradigm has proven highly effective, achieving strong empirical success across diverse offline RL and robotic control benchmarks. Despite this success, since the guidance module and the diffusion model are trained jointly and tightly coupled in inference, TWO CHALLENGES exist: 1)Before the guidance module has converged, it cannot provide effective guidance to the diffusion model or even mislead the training process (Kim et al., 2023). 2)This tightly coupled diffusion and guidance usage prevents re-usage across different combinations of network modules.

Existing methods solve CHALLENGE 1 by pretraining the diffusion model and then letting the diffusion model generate samples for the guidance training. Existing methods pretrain the diffusion model and rely on synthetic samples for guidance training. This design introduces two limitations: (1) generated samples may

inherit and amplify modeling biases due to distribution shift, and (2) the diffusion backbone is optimized only for empirical distribution matching rather than reward alignment. In this work, we take the opposite approach: pre-training the guidance module, then freezing it to guide the diffusion model, which is called Guidance-First Diffusion Training (GFDT). This design offers three parallel advantages. 1)Because the entire model is trained solely on the dataset, without introducing synthetic samples that could contaminate the offline data. (As a purely offline RL formulation, the method naturally does not involve exploration out of the dataset.) 2) Unlike the existing pipelines that train the diffusion model to generate mixed-quality samples, GFDT updates diffusion using reward-aligned gradient shifts. This converged guidance module ensures every generated action is already value-optimized rather than unconstrained. 3) Since diffusion learning is guided by a well-trained guidance signal, optimization converges substantially faster by avoiding unstable early-stage guidance updates.

To solve CHALLENGE 2, CFG methods can be modularized: Further, we cross-combined components from models trained under different frameworks. Specifically, we took the guidance module from an IDQL model and paired it with a completely separate DQL diffusion model—one that had never been trained in conjunction with the IDQL guidance. The plug-and-play (PAP) module and its reversed model are functional, and the performance is comparable with the stronger of these two, and has a significantly better early training stage, outperforming both the baselines. This result strongly suggests that the relationship between the guidance module and the diffusion model is modular and flexible, and therefore, highlights a promising direction for future research.

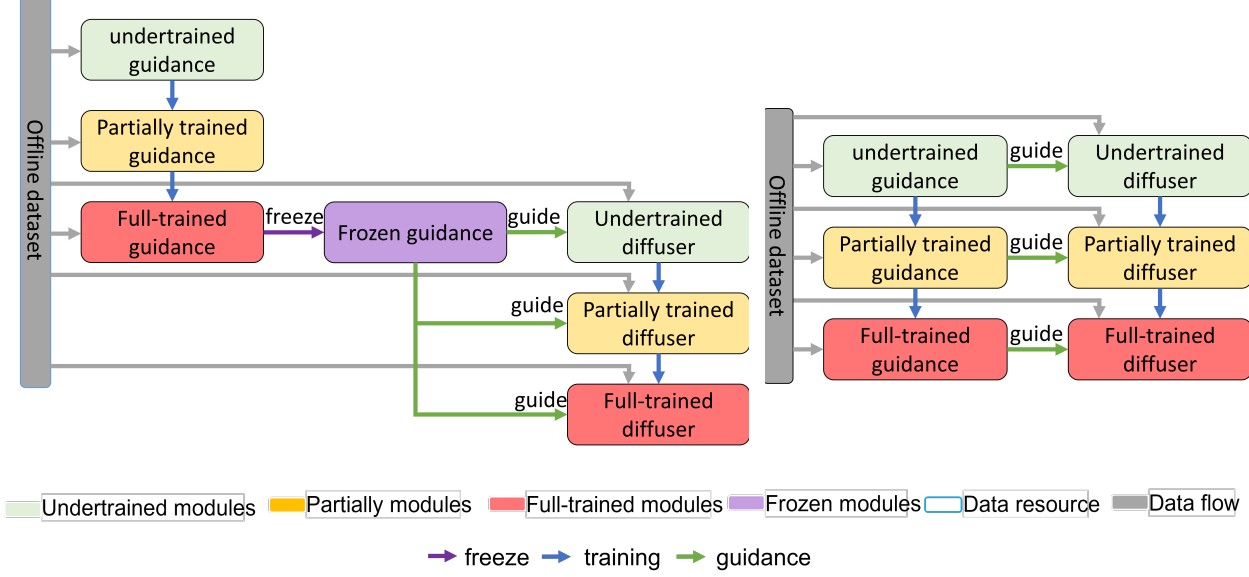

Figure 1: **Schematic Illustration of GFDT versus Traditional CFG.** GFDT (left) first trains and freezes the guidance module before diffusion training, ensuring that diffusion is guided only by a converged and reliable guidance signal. In contrast, traditional CFG (right) jointly trains guidance and diffusion, during which diffusion is influenced by immature guidance for a substantial portion of training. The illustration is conceptual and highlights training dynamics; labels such as "undertrained" and "partially trained" are qualitative for visualization purposes.

---

**Algorithm 1** Guidance-First Diffusion Training (GDFT)

---

**Require:** Offline dataset $\mathcal{D} = \{(s, a, r, s')\}$, Q-network $Q_\phi$, diffusion model $\epsilon_\theta$, guidance scale $\lambda$, training steps $N_q$, $N_\theta$

1: **// Step 1: Train Q-function (guidance model)**
2: **for** $i = 1$ to $N_q$ **do**
3:      Sample minibatch $(s, a, r, s') \sim \mathcal{D}$
4:      TD target: $y \leftarrow r + \gamma \cdot \max_{a'} Q_\phi(s', a')$
5:      Update $Q_\phi$ to minimize $\mathcal{L}_Q = \|Q_\phi(s, a) - y\|^2$
6: Freeze $Q_\phi$
7: **// Step 2: Train diffusion model with frozen guidance**
8: **for** $j = 1$ to $N_\theta$ **do**
9:      Sample $(s, a_0) \sim \mathcal{D}$
10:      Add noise: $a_t \leftarrow \sqrt{\bar{\alpha}_t} a_0 + \sqrt{1 - \bar{\alpha}_t} \cdot \epsilon$, $\epsilon \sim \mathcal{N}(0, I)$
11:      Predict noise: $\hat{\epsilon} \leftarrow \epsilon_\theta(a_t, s, t)$
12:      Update $\epsilon_\theta$ to minimize $\mathcal{L}_{\text{diff}} = \|\hat{\epsilon} - \epsilon\|^2 + \mathcal{L}_Q$
                                                   $\triangleright$ $Q_\phi$ frozen; $\mathcal{L}_Q$ still updates $\epsilon_\theta$

13: **Return** trained $\epsilon_\theta$, frozen $Q_\phi$
14:
15: **Inference:**
16: **for** denoise steps **do**
17:      Sample candidate $a_k \sim \pi_\theta(\cdot | s)$
18:      Apply guidance: $a_{k-1} \leftarrow a_k + \lambda \cdot \nabla_a Q_\phi(s, a_k)$

---

The detailed pseudo-codes the traditional CFG, and a diagram comparison can be found in Appendix A.1. In prior works, sometimes diffusion-based policies are interpreted as actors, and Q-based guidance modules are interpreted as critics. In this paper, we use the terms Diffusion module and Guidance Module to emphasize that our method relies on diffusion dynamics rather than generic Actor-Critic training.

## 2 Preliminaries

In offline RL, the objective is to learn a policy $\pi_\theta(a|s)$ that maximizes the expected return using only a fixed dataset $D = \{s, a, r, s'\}$, with interaction to the environment only at the inference stage (Levine et al., 2020). When diffusion models are used as policies, several offline RL methods—including Diffusion Q-Learning (DQL) (Yang et al., 2023), Exponential Diffusion Process (EDP) (Kang et al., 2023), and Implicit Diffusion Q-Learning (IDQL) (Hansen-Estruch et al., 2023)—adopt a *conditional denoising diffusion process*. The policy is defined implicitly by the reverse denoising procedure: $\hat{a}_0 \sim \pi_\theta(\cdot \mid s)$.

**Q-Guided Diffusion:** Intuitively, the diffusion model remains the policy backbone, while the Q-function only provides a directional bias to the denoising process. Q-Guided Diffusion(Janner et al., 2022) has a forward noising process and a reverse denoising process. The forward noising process gradually perturbs a clean action $a_0$ into a noisy version $a_k$:

$$a_k = \sqrt{\bar{\alpha}_k} \, a_0 + \sqrt{1 - \bar{\alpha}_k} \, \epsilon, \quad \epsilon \sim \mathcal{N}(0, I), \quad \bar{\alpha}_k = \prod_{i=1}^{k} \alpha_i, \tag{1}$$

where $k \in \{1, \dots, T\}$ denotes the diffusion timestep (or noise level), and $\bar{\alpha}_k$ is the cumulative product of noise scheduling coefficients . A denoising network $\epsilon_\theta(a_k, k, s)$ is trained to estimate the clean action $a_0$ from its noisy counterpart $a_k \sim \mathcal{N}(0, I)$. During reverse diffusion, we can form an estimate of the clean action:

$$\hat{a}_{k-1} = \frac{1}{\sqrt{\bar{\alpha}_k}} \Big( a_k - \sqrt{1 - \bar{\alpha}_k} \cdot \epsilon_\theta(a_k, k, s) \Big). \tag{2}$$

Rather than optimizing a specific clean action directly, the objective of the denoising is to learn a denoising network $\epsilon_\theta$ that implicitly defines a policy capable of generating $a_0$ through reverse diffusion. This part of $\epsilon_\theta$ training is formulated by minimizing the behavior clone loss is formulated as: $\mathcal{L}_{\text{BC}} = \mathbb{E}_{a_0, \epsilon, t}[\|\epsilon_\theta(a_t, t, s) - \epsilon\|^2]$

plus the reward signal $\mathcal{L}_Q$, and the entire loss becomes: $\mathcal{L}_{\text{actor}} = \mathcal{L}_{\text{BC}} + \eta \, \mathcal{L}_Q$. ($\eta$ is a trade-off coefficient that balances behavior cloning against Q-guidance.)

Behavior cloning($\mathcal{L}_{\text{BC}}$) provides the base diffusion policy, while main stream Q Guided policies incorporate a learned Q-function as a value guidance signal to bias the denoising direction toward higher-return actions. Here $Q_\phi(s,a)$ denotes the learned Q-function, i.e., an estimation of the expected discounted return starting from state $s$ and action $a$: $Q_\varphi(s,a) \approx \mathbb{E}[\sum_{t=0}^{\infty} \gamma r_t \,\big|\, s_0 = s, a_0 = a]$, ($\gamma$ is the discount factor, $r_t$ the reward at step $t$). For stable guidance, we normalize this value by its expectation over sampled actions, defining $\tilde{Q}_\phi(s,a) = Q_\phi(s,a)/\mathbb{E}_a[Q_\phi(s,a)]$, which ensures consistent guidance magnitude across methods. Building on this unified formulation, the following methods can be viewed as variations of the Q-guided diffusion paradigm.

**DQL**: In the DQL instantiation considered here, value guidance is implemented with a double-Q value estimator, taking the minimum of two Q-values to improve estimation reliability and mitigate overestimation bias (van Hasselt, 2010).

$$\mathcal{L}_\pi^{\text{DQL}}(\theta) = -\mathbb{E}_{s\sim\mathcal{D}} \left[\min\left(Q_{\phi_1}(s,a_0),\, Q_{\phi_2}(s,a_0)\right)\right], \quad \text{where } a_0 \sim \pi_\theta(\cdot \mid s). \tag{3}$$

At inference time, every step after $a_k$ is obtained by denoising an earlier step as described in Equation2, the learned Q-function provides explicit guidance by adjusting the sampled actions:

$$\tilde{a}_{k-1} = \hat{a}_{k-1} + \lambda \frac{\nabla_a Q_\phi(s, \hat{a}_{k-1})}{\|\nabla_a Q_\phi(s, \hat{a}_{k-1})\| + \delta}, \tag{4}$$

where $\lambda$ balances exploitation of the learned Q-function against fidelity to the diffusion process. This guidance follows standard diffusion-style normalization practices (Ho et al., 2020) and $\delta$ is a small constant added for numerical stability.

**IDQL:** Unlike DQL, which directly incorporates the reward target into the diffusion training, IDQL adopts a two-stage design. During training, the diffusion model is updated purely via behavior cloning from the dataset, without any explicit Q-guidance. In parallel, a separate Q-value network is learned to estimate $Q(s,a)$. At inference time, the diffusion model offers candidate actions, and the guidance module evaluates them with the Q-network through a one-hot encoding scheme, and selects the action with the highest estimated Q-value. Compared to traditional Q-guided diffusion, IDQL thus applies the guidance only to the inference stage rather than the training stage. Although IDQL does not leverage the Q-value estimator to directly guide diffusion during training, the implementation still places the Q-network update within the same training loop as the diffusion model. This coupling is not theoretically necessary, and the consequence is a substantially higher computational and memory usage. There is a detailed analysis of the difference between IDQL and GFDT in Appendix A.2. Since IDQL does not employ diffusion guidance during training, it tends to underperform on dense-reward tasks that require fine-grained, step-wise policy optimization. In contrast, it performs well on in-distribution, goal-oriented environments such as AntMaze, which are evaluated based on task success rather than on the efficiency or smoothness of the trajectory.

**EDP:** EDP introduces a key modification on traditional Q-guided diffusion: *one-step denoising*, instead of running the full reverse chain to generate actions. EDP corrupts a dataset action $a^0$ to $a^k$ in one step, and then one-step-reconstructs an approximate clean action $\hat{a}^0$ using the denoise backward path as Equation 6. This *action approximation* replaces expensive sampling with a lightweight inference step, making EDP orders of magnitude faster to train.

In traditional diffusion policies, generating a clean action $a^0$ requires running a long reverse chain:

$$a^T \to a^{T-1} \to \cdots \to a^0,$$

where $T$ is typically large (e.g., 100–1000). This iterative procedure is computationally expensive.

Instead, EDP corrupts a dataset action $a^0$ directly into a noisy action $a^k$ in one step:

$$a^k = \sqrt{\bar{\alpha}_k}\, a^0 + \sqrt{1 - \bar{\alpha}_k}\, \epsilon, \quad \epsilon \sim \mathcal{N}(0, I). \tag{5}$$

Then, it reconstructs an approximate clean action $\hat{a}^0$ by applying the denoiser once:

$$\hat{a}^0 = \frac{1}{\sqrt{\bar{\alpha}_k}}\, a^k - \frac{\sqrt{1 - \bar{\alpha}_k}}{\sqrt{\bar{\alpha}_k}} \cdot \epsilon_\theta(a^k, k, s), \tag{6}$$

where $\epsilon_\theta$ is the learned denoising network conditioned on state $s$.

These three methods are the baselines of our methods. Currently, our modifications are typically only applied to Temporal Difference (TD) based methods and cannot be applied to Trajectory-Based methods(e.g., (Janner et al., 2022; Ajay et al., 2023)) as they use return annotations for full sequences $\tau = (s_0, a_0, ..., s_T)$. Experiments showed low convergence rates and suboptimal precisions in their Q-value predictions, and therefore, Guidance-First Diffusion Training showed little improvement. We attribute this observation to the following factors: when the trajectory is long, the return may be noisy due to stochastic behavior policies, so reward attribution over entire sequences can be ambiguous. (It is hard to infer which action caused the reward change.) In contrast, TD-based methods that work on $(s, a)$ pairs avoid this issue and support more granular, local learning signals.

## 3 Methodology

Having introduced the GFDT method and modular module composition in the introduction, we now extend the discussion with a mathematical model to justify the approach. First, we explain why a pretrained, converged model can provide more accurate guidance to the diffusion model, better than an unconverged model that is trained jointly with it, thereby accelerating the training process. Second, we prove that as long as the guidance model can provide a direction that leads to reward improvement and the step size is small enough, it is sufficient for guiding the diffusion model, even if the guidance model was not co-trained with the diffusion model.

### 3.1 Guidance Dynamics and Convergence

**Theoretical Motivation.** We build upon the theoretical framework established by Theorem (Fujimoto et al., 2019a), which introduces Batch-Constrained Q-Learning (BCQL) as a value-based method with provable convergence guarantees under offline settings. *Given a deterministic Markov Decision Process (MDP) and coherent batch $\mathcal{B}$, along with standard Robbins-Monro convergence conditions on the learning rate, BCQL converges to $Q^{\pi_\mathcal{B}}(s, a)$, where $\pi^*(s) = \arg\max_{a\ s.t.\ (s,a)\in\mathcal{B}} Q^{\pi_\mathcal{B}}(s, a)$. This policy is guaranteed to match or outperform any behavioral policy contained in the dataset.*

**Batch-Constrained Guidance via Diffusion.** Inspired by this result, we adopt a *batch-constrained guidance* framework for training diffusion policies. Our key design choice is to pretrain a guidance policy (a value function $Q_p(s, a)$ or behavioral pr) purely on the offline dataset, then use it to train and guide the sampling of a diffusion policy. The above theorem guarantees, the training on the offline dataset converges to an optimal reward estimation policy Q*(s,a). Note that the $Q_\phi$ in the Preliminaries refers to a general-purpose value estimator, whereas here $Q_p$ denotes a pre-trained guidance policy learned from offline data.

This optimized $Q_p(s, a)$ is then added to the gradient update of the training process of the diffusion:

$$\nabla_\theta \mathcal{L}_{\text{total}} = \nabla_\theta \mathcal{L}_{\text{BC}} + \eta \nabla_\theta Q_p(s, a), \tag{7}$$

where $\mathcal{L}_{\text{BC}}$ is a behavior cloning loss that regularizes the learned policy to stay close to the empirical distribution of $\mathcal{B}$, and $\eta$ controls the strength of the guidance. Both of these two parts are learned based on the same dataset $\mathcal{B}$, so term 1 offers regularization to avoid leaving the support of the offline dataset and term 2 encourages reward optimization.

To further ensure batch-constrained optimization behavior during action generation(inference step), we modify the reverse diffusion process by injecting $Q_p$ gradients of the pretrained module:

$$a_{t-1} \leftarrow f_{\text{diffusion}}(a_t, \hat{a}_0, t) + \lambda \frac{\nabla_a Q_p(s, \hat{a}_t)}{\|\nabla_a Q_p(s, \hat{a}_t)\| + \delta} + \sqrt{2\tau_t}\,\xi_t, \tag{8}$$

where $\lambda$ is a guidance coefficient. This operation encourages the final action $a_0$ to lie in high-reward regions. Importantly, because the value function $Q(s, a)$ is trained on the offline dataset $\mathcal{B}$, and the diffusion model itself models the same data distribution. Intuitively, in every inference step, the diffusion model generates an action, and the Q module shifts the gradient in a magnitude that is close to $\lambda$. So, the generated action must be inside or close to the region of the dataset $\mathcal{B}$. Thus, the value-aware perturbation can be interpreted as a form of approximate, soft batch-constrained policy improvement. For a more detailed explanation, please

check Appendix D. On the contrary, before sufficient training, the inaccurate gradient will misshape the diffusion distribution and prolong the training process by adding incorrect information to the optimization, which may not be corrected in later training.

Admittedly, our method does not strictly satisfy all assumptions of BCQL. Under ideal settings, directly optimizing the Bellman objective admits a unique fixed point and converges to the optimal value function. In practice, however, we approximate the value estimator with a neural network, which leads to a non-convex optimization problem. As a result, training may converge to a local minimum rather than the true Bellman optimum, which is difficult to avoid in real-world implementations. Nevertheless, our framework inherits the core principle of batch-constrained learning: value-based guidance should be sufficiently trained within the data distribution so that gradient guidance remains reliable. This interpretation is consistent with our empirical results (Section 4), where in-distribution guidance improves the performance of CFG. Conversely, the analysis in Section 5 shows that out-of-distribution guidance often produces unstable gradients and degrades performance in practice.

## 3.2 Decoupling Guidance from Diffusion Training

**Proposition 2.** *The effectiveness of guidance does not depend on whether the Q-function is jointly trained with the diffusion policy. Instead, the key requirement is that the guidance signal provides a consistently positive and smooth directional influence on action updates. Therefore, external Q-function that yields beneficial value gradients can be modularly integrated to steer the diffusion process.*

Let $\pi_\theta$ be a diffusion policy and $Q_\phi$ a value function, both trained on dataset $\mathcal{D}$. The guided sampling update is:

$$a_{t+1} = a_t + \alpha_t \nabla \log \pi_\theta(a_t|s) + \lambda \frac{\nabla_a Q_\phi(s, a_t)}{\|\nabla_a Q_\phi(s, a_t)\|} + \sqrt{2\tau_t}\xi_t. \tag{9}$$

By expansion, the expected one-step gain from guidance is $\lambda\|\nabla_a Q_\phi(s, a_t)\|$, independent of $\pi_\theta$. This reveals a key structural property: the score term $\nabla \log \pi_\theta$ enforces adherence to the data manifold, while the gradient term $\nabla Q_\phi$ biases sampling toward high-reward regions. These two functions are *intrinsically decoupled*—they operate on different objectives and require no joint adaptation.

Consequently, a guidance module trained on $\mathcal{D}$ can be combined with a diffusion model also trained on $\mathcal{D}$ without performance degradation, explaining our plug-and-play results (Section 4.2). The only requirement is that $\lambda$ remains small enough to keep samples within $\mathrm{supp}(\mathcal{D})$—a standard batch-constrained condition in offline RL. When guidance is trained on a different distribution, this condition breaks, leading to the out-of-distribution failures documented in Section 5.

## 3.3 Acceleration through Value Guidance

**Heuristic sample-complexity argument.** To connect our theoretical discussion to practical training savings, we provide a heuristic estimate of how many gradient steps can be saved. A standard result in nonparametric estimation theory states that estimating the gradient of a smooth function from $N$ i.i.d. samples achieves an error that decays as $O(1/\sqrt{N})$ (Wasserman, 2006; Stone, 1982). For a Lipschitz function class with constant $L$, a guidance model $Q_\phi$ trained to accuracy on dataset $\mathcal{B}$, the gradient estimation error satisfies:

$$\mathbb{E}\|\nabla_a Q_\phi - \nabla_a Q^*\| \leq C * \frac{L}{\sqrt{N}} + \epsilon \tag{10}$$

Table 1: The parameters used in Equation (10)

| $\nabla_a Q_\phi$ | Pretrained Guidance Gradient | $\epsilon$ | Final training error of $Q_\phi$ |
|---|---|---|---|
| $L$ | Lipschitz constant of the Q-function class | $N$ | Number of samples in dataset $\mathcal{B}$ |
| $\nabla_a Q^*$ | True gradient of the optimal Q-function | $C$ | a constant depends on the underlying function class |

Heuristically, although in RL practice the Lipschitz constant $L$ is rarely specified, in related areas (e.g., WGAN with spectral normalization (Miyato et al., 2018)) one often enforces $L = 1$. Under this convention,

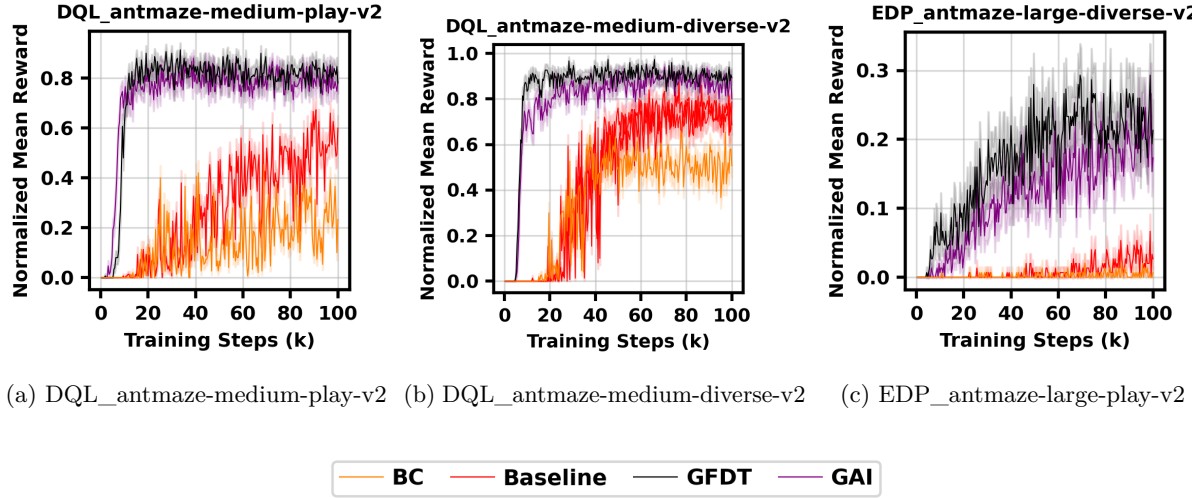

(a) DQL_antmaze-medium-play-v2  (b) DQL_antmaze-medium-diverse-v2  (c) EDP_antmaze-large-play-v2

Figure 2: **Training performance of proposed algorithms on benchmarks.**

Eq (10) implies that achieving a guidance accuracy of $\mathbb{E}\|\nabla_a Q_\phi - \nabla_a Q^*\| = 0.01$ would require about $N = 1/\mathbb{E}\|\nabla_a Q_\phi - \nabla_a Q^*\|^2 = 10^4$ effective samples (equivalently, $\sim 10^4$ gradient updates of the Q network). In more parameter-sensitive scenarios, pushing the accuracy further to $\mathbb{E}\|\nabla_a Q_\phi - \nabla_a Q^*\| = 0.001$ would require as many as $N = 10^6$ effective samples (i.e., $10^6$ gradient updates). This illustrates the significant optimization steps that are potentially misguided before the guidance converges, and shows how our method helps reduce otherwise wasted early training of the diffusion model. Pretraining is therefore essential to ensure that guidance has semantic meaning before being applied in generation. Appendix. I shows the detailed calculations.

## 4 Experiments

We evaluate our method on standard tasks from the PyBullet D4RL benchmark (Fu et al., 2021; Foundation, 2024). To ensure fair comparison, we re-train three seeds of three representative diffusion-based offline RL algorithms—EDP, DQL, and IDQL(module interchange)—based on the dataset and benchmark code from (Wang et al., 2024). These methods serve as illustrative examples of our theory, which can be easily extended to more diffusion methods. For more details about the environment setting, please check Appendix C. The detail comparison between our model and the baseline models is shown in Appendix E. All experiments were conducted using PyTorch on a high-performance computing (HPC) cluster. We adhered to any assigned framework for our work because the focus of this paper is not restricted to any specific computational framework.

### 4.1 Performance of GFDT

.

As shown in the Table 2, GFDT enables diffusion-based policies to achieve **comprehensive superiority** over established mainstream offline RL algorithms. DQL-GFDT outperforms all non-diffusion baselines on 11 out of 13 tasks, elevating average performance from **86.9** to **109.3**. EDP-GFDT, despite its single-step denoising architecture, consistently matches or exceeds these strong baselines with an average of **96.4**. Furthermore, our plug-and-play configurations demonstrate the modularity of our approach: DGID (DQL-guidance + IDQL-diffusion) achieves **96.7**, while IGDD (IDQL-guidance + DQL-diffusion) reaches **99.4**, both surpassing all non-diffusion baselines despite never being co-trained. These results establish GFDT as a new performance standard for diffusion-based offline RL.

Table 2: Comparison with standard offline RL baselines on D4RL tasks. Results ReBR, short for REBRAC from (Wu et al., 2019) DICE (Ma et al., 2024), CQL are taken from (Kumar et al., 2020), IQL from (Kostrikov et al., 2022), EDP-GFDT and DQL-GFDT, DGID and IGDD are ours. (These abbreviations are in the Appendix. H)

| Env | ReBR | DICE | CQL | IQL | DQL_GFDT | EDP_GFDT | DGID | IGDD |
|---|---|---|---|---|---|---|---|---|
| HCME | 101.1±1.5 | 97.3±0.6 | 45.3±0.3 | 92.7±2.8 | 90.2±0.4 | 87.2±0.0 | 85.61±0.01 | 84.77±0.00 |
| HCMR | 51.0±0.2 | 49.2±0.9 | 45.3±0.3 | 42.1±3.6 | **67.9±0.3** | **64.4±0.0** | **70.6±0.8** | **69.8±0.0** |
| HCMV | 65.6±1.0 | 60.0±0.6 | 46.9±0.4 | 50.0±0.2 | **67.0±0.6** | 59.3±0.4 | 60.9±0.9 | 60.9±0.2 |
| HOME | 107.0±6.4 | 112.2±0.3 | 96.6±2.6 | 85.5±29.7 | **172.3±1.1** | **163.9±0.0** | **182.4±0.0** | **182.7±0.0** |
| HOMR | 98.1±1.6 | 102.3±2.1 | 89.6±13.2 | 89.6±13.2 | **153.0±0.8** | **146.4±1.7** | **157.2±0.0** | **157.2±0.0** |
| HOMV | 102.0±1.0 | 100.2±3.2 | 61.9±6.4 | 65.2±4.2 | **147.0±1.1** | **141.6±0.0** | **136.5±0.2** | **136.5±0.2** |
| WAME | 111.6±0.3 | 114.1±0.5 | 104.6±1.1 | 112.1±0.5 | **117.6±0.3** | **116.6±0.0** | **118.1±0.0** | **118.4±0.0** |
| WAMR | 77.3±7.9 | 90.8±2.6 | 76.8±1.0 | 75.4±9.3 | **95.6±4.8** | 83.8±0.4 | **93.1±0.94** | 87.5±0.7 |
| WAMV | 82.5±3.6 | 89.3±1.3 | 79.5±3.2 | 80.7±3.4 | **87.9±0.3** | 84.2±0.0 | 87.11±0.07 | 87.1±0.0 |
| L-div | 54.4±25.1 | 91.3±3.1 | 14.9 | 27.6±7.8 | 90.7±5.4 | 31.3±6.9 | 33.3±6.9 | 46.7±7.0 |
| L-play | 60.4±26.1 | 85.7±4.8 | 15.8 | 42.5±6.5 | **89.3±5.6** | 22.7±6.7 | 28.0±6.7 | 59.3±7.0 |
| M-div | 76.3±13.5 | 68.6±8.6 | 53.7 | 61.7±6.1 | **97.3±4.0** | 52.0±7.7 | 58.0±7.0 | 68.0±6.8 |
| M-play | 84.0±4.2 | 72.0±6.5 | 61.2 | 64.6±4.9 | **91.3±5.3** | **118.0±10.4** | 56.7±7.0 | 74.7±6.6 |
| Average | 84.6 | 86.9 | 58.7 | 69.2 | **109.3** | **96.4** | **96.7** | **99.4** |

*Note 1* While the algorithms were proposed in the aforementioned papers, the authors did not release their raw experimental results;all numerical values for standard baselines in Table 2 are taken from (Ma et al., 2023). These methods are considered as the strongest methods in offline RL.

*Note 2* The left four columns show results from other methods, while the right four columns present results from our proposed method. The highest values in the left four columns are highlighted in blue. If a value in our method (right columns) exceeds the highest value on the left, it is shown in bold with a gray background.

Table 3: Ablation study results (averaged across environments)

| Domain | Algorithm | Baseline | GFDT | GAI | BC | Unfreeze |
|---|---|---|---|---|---|---|
| MuJoCo | DQL | 112.47 ± 0.98 | **115.18 ± 0.92** | 99.50 ± 10.47 | 103.34 ± 8.10 | 110.92 ± 7.57 |
| | EDP | 105.94 ± 0.25 | **108.96 ± 0.63** | 91.59 ± 3.17 | 103.34 ± 8.10 | 106.92 ± 0.99 |
| AntMaze | DQL | 78.50 ± 37.51 | **92.17 ± 5.07** | 89.50 ± 5.49 | 65.67 ± 6.80 | 91.83 ± 5.14 |
| | EDP | 48.16 ± 8.20 | **56.00 ± 7.92** | 34.83 ± 49.26 | 30.83 ± 6.89 | 49.83 ± 7.52 |

As detailed in Appendix E (Tables 11 and 12), GFDT consistently outperforms its non-guided counterparts across the majority of tasks: DQL-GFDT surpasses DQL-baseline in 11 out of 13 environments, while EDP-GFDT outperforms EDP-baseline in 9 out of 13 environments. The performance gains are particularly pronounced in challenging sparse-reward AntMaze tasks, where GFDT delivers substantial improvements of 14-20%. This demonstrates that Guidance-First training is especially effective in goal-directed environments where precise credit assignment is critical—precisely the settings where stable, converged guidance signals matter most.

### 4.1.1 The role of the guidance module

To analyze the role of reward guidance and whether the guidance is removable or replaceable, we conducted an ablation study of a series of training sessions. In this experiment, we compare the performance of GFDT, baseline models, an ablation study that removed the reward guidance part of the baseline training—behavior clone(BC), an ablation study on a behavior clone training with a pretrained guidance at the inference stage(GAI), an ablation study that pretrained the guidance module but did not freeze it in diffusion training.

Beyond the superior performance of our proposed algorithm, we further investigate its other properties through the ablation Study. The pretrained but non-frozen guidance model in diffusion training outperforms the

Table 4: Training gradient steps required to reach 95% performance (batch size 256). Lower is better. Values in parentheses indicate relative cost compared to baseline.

| Domain | DQL | GFDT_DQL | EDP | GFDT_EDP | DDIG | IDDG |
|--------|-----|----------|-----|----------|------|------|
| MuJoCo | 37733 | 14233 (34.40%) | 29500 | 22533 (72.34%) | 35967 (65.86%) | 28667 (59.37%) |
| AntMaze | 847400 | 27500 (7.48%) | 1275000 | 601200 (41.28%) | 66000 (29.11%) | 139600 (34.16%) |

Table 5: Parameter statistics summary for different model widths. (The applied model has a hidden layer size of 256.)

| | 128 | 256 | 512 |
|--|-----|-----|-----|
| Diffusion Params | 172,230 (70.0%) | 174,534 (38.1%) | 172,230 (13.7%) |
| Guidance Params | 73,986 (30.0%) | 283,650 (61.9%) | 1,082,370 (86.3%) |
| Total Memory | 0.94 MB | 1.75 MB | 4.79 MB |

baseline but does not surpass GFDT, confirming that additional training of the guidance model is unnecessary. The slight performance drop is likely caused by overfitting of the guidance model under over-training. BC and GAI, removing the guidance module(in training), results got worse performance, underscoring the important role of guidance. This supports our claim that a valid guidance model is critical for successful training.

The training time of the diffusion model decreased significantly as shown in Table 4 and Table13 with decreased computational resources—while they have the same training for the guidance module, the training steps for the diffusion module was significantly decreased. The following sections discuss how to plug and play a pretrained module into the diffusion model, making the modules reusable.

In Table 5, we analyze the number of parameters in the guidance module and the diffusion model. The parameter counts objectively reflect the memory footprint and computational cost of each component. Since we pre-train the guidance module, the peak memory usage during GFDT training only needs to accommodate the largest of these components. Therefore, we consider the effective reduction to be determined by the smaller of the two overlapping components, which corresponds to a 38.1% decrease for our model.

### 4.1.2 Ablation Study on Guidance Maturity and Final Performance

**Ablation Study on Guidance Maturity.** we conduct a set of ablation studies to examine the interaction between guidance maturity and final performance. Specifically, we compare three training strategies: (1) a fully pre-trained guidance module (2) a jointly trained guidance module without pre-training(baseline), and (3) two intentionally delayed settings in which the convergence speed of the guidance module is slowed down by $100\times$ and $1000\times$, respectively, forcing the guidance signal to mature significantly later. Overall, the pre-trained guidance consistently achieves stronger final performance than the baseline, in line with our previous observations. Notably, when the maturity gap is further enlarged through slowed training, a clear performance degradation is observed. In particular, the $100\times$ and $1000\times$ delayed settings lead to performance drops of approximately 8%–38% across different tasks. These results suggest that accurate guidance during the early phase of training plays an important role in shaping optimization trajectories. When the guidance signal emerges only at later stages, its influence appears limited once the model has already converged toward suboptimal regions. Taken together, the findings provide additional evidence that early-stage guidance accuracy is a key factor underlying the performance gains of GFDT. The summary of the ablation study is show in Table 6 and Table 7 and a detailed result is show in Table 14 and Table 15.

**Quality-Level Ablation of the Guidance Module.** In addition to maturity speed, we further investigate how the quality of the guidance module affects training outcomes. We construct three variants GFDT: (1) a high-quality guidance module obtained after full convergence (Real GFDT), (2) The overall performance when a a half-trained guidance module is used as the pretrained framework(half trained GFDT) and (3) a untrained initialized guidance module(untrained GFDT). The results demonstrate that guidance quality plays a decisive role. Using the fully trained guidance module provides strong and stable improvements, whereas the

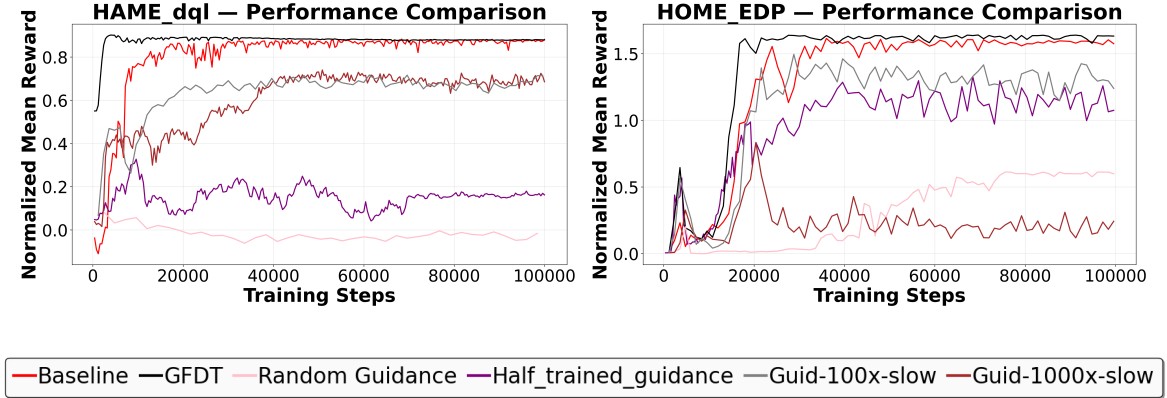

Figure 3: Performance comparison under different guidance maturity speeds. Pre-trained guidance achieves stable improvements, while delayed guidance leads to persistent performance degradation.

Table 6: Ablation study on slow guidance (Average only)

| Method | baseline | GFDT | 100xslow | 1000xslow |
| --- | --- | --- | --- | --- |
| DQL | 112.47 | 115.18 | 103.8550 | 69.9917 |
| EDP | 105.94 | 108.96 | 92.2510 | 81.0894 |

Table 7: Ablation study on half-trained guidance (Average only)

| Method | baseline | GFDT | half-trained | RAN |
| --- | --- | --- | --- | --- |
| DQL | 112.47 | 115.18 | 45.1693 | 17.5713 |
| EDP | 105.94 | 108.96 | 73.2580 | 32.2449 |

half-trained guidance leads to noticeably worse performance. However, its consistent outperforming untrained guidance. This result is to demonstrate the GFDT framework needs high quality pretrained guidance.

## 4.2 Plug-and-Play Model Composition

Our research reveals that diffusion models exhibit unique plug-and-play compatibility with their guidance modules, following the theoretical proof. We evaluate two hybrid configurations without any additional training: 1) DQL-as-Guidance + IDQL-as-Diffusion(IDDG), 2) IDQL-as-Guidance + DQL-as-Diffusion(DDIG). as shown in Figure 4, both combinations has the following properties. (1) They achieved final performance comparable to the DQL baseline, the higher one of these two models in Mujoco environments. (2) They performed lower results in Antmaze environments because the Plug and Play method does not match with the Max Q algorithm that is applied in Antmaze environments (3) They exhibited significant faster initial convergence speeds compared to baselines; and (4) they maintain stability despite architectural misplacement. The detailed analysis of this ablation study is in the Appendix.F. The key point is that for environments with dense reward (Mujoco), the discrepancy or mismatch can be compensated and corrected immediately, the module interchange is applicable, and even beneficial for optimality and for smoothing the adjustments. However, for sparse rewarding environments(Antmaze) that need high precision, the error can accumulate, and the reward estimation becomes out-of-distribution, because of the mismatch.

Most importantly, despite the absence of joint training or any task-specific fine-tuning, the plug-and-play configurations achieve performance comparable to established baselines such as CQL and IQL. Although their results are slightly below the best-performing methods (e.g., DICE), the fact that two independently trained modules can be directly combined to reach this level of performance highlights the strong modular compatibility and generalization potential of our framework. This result implies that: Guidance modules can provide effective policy improvement signals regardless of diffusion model architecture with proper setup. It implies that the normalizing step in the training and inference parts is expected to be an important reason why these two modules are interchangeable, as explained in the Methodology Section. Finally, early-stage advantages suggest that when the modules are both under-trained and are not perfect, a differently structured model is not only unharmful, but can cross-correct with the errors of each module. We find that a guidance module—trained independently and even with a completely different architecture—can be directly applied to

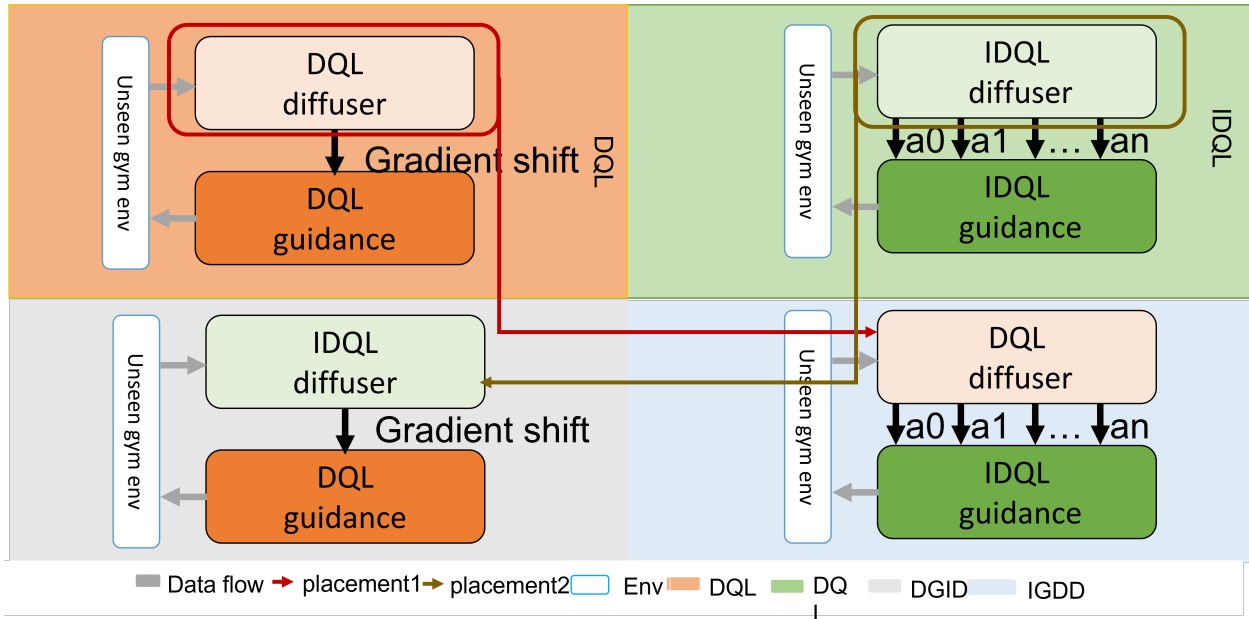

Figure 4: Diagram of plug-and-play modular configuration

a diffusion model. This experiment is consistent with our theoretical analysis that, as long as the guidance model provides a reliable estimate of the reward(gradient estimate per se), it can be reused across structurally dissimilar models. Such modularity enables flexible training pipelines and paves the way for reusable, task-agnostic RL components. Finally, as we have mentioned in the theoretical part, if the guidance model is not independent trained on the same data distribution, there is no guarantee that the independent module will be beneficial. In the following section, we illustrate that pretraining on out-of-distribution(OOD) data is more likely to be detrimental than beneficial in practice.

## 5 Application scope and Limitations

To illustrate the distribution shift between datasets, We performed PCA analysis on the datasets and plotted the region that contains 60% of the data points to eliminate the instability caused by outliers. The highlighted points are the top 100 highest-reward points. The positions of these high-reward points varied significantly across datasets, even within the same environment and the shifting trade suggests the gradual converging process. *The differences in the plot imply that for any given dataset, the data from another expertise level often lies in an OOD region, as shown in Table.5.* The dataset distributions of other environments are show in Appendix G. Based on this observation, we conducted OOD experiments by training a guidance network on one dataset, freezing it, and then using it to guide the training of a main diffusion model on a different dataset. Across 36 tested models, 28/36(78%) exhibited significant performance degradation. Among the eight cases with good performance, half benefited from guidance modules pretrained using medium replay models, which cover a broader range of data and thus generalize better than other datasets.

From these results, we conclude that the decouple method is highly sensitive to distribution shift. Training a guidance network on in-distribution data is critical for success. Otherwise, even expert-level data cannot reliably guide training on a different distribution. This ablation study is consistent with the conclusion in the Methodology part that OOD pretraining cannot guarantee performance improvement.

Another kind of limitation of Q-guided diffusion, not only in our methods but also, is whether the guidance should exist in the training. As shown in Table 9, for high-dimensional tasks with narrow data coverage (e.g., Adroit), even slight guidance can push the policy out-of-distribution, causing performance to drop significantly. In these cases, relying solely on behavioral cloning proves more stable and effective. Conversely, for tasks with moderate difficulty and abundant data, Q-guidance can successfully enhance performance.

Table 8: Performance comparison with standard deviation in ± format (2 decimal places)

| Dataset | DQL | IDQL | DGID | IGDD |
|---|---|---|---|---|
| halfcheetah-expert | 88.21±0.28 | 71.07±0.02 | 85.59±6.92 | 81.64±0.86 |
| halfcheetah-medium-expert | 88.28±0.53 | 72.03±0.01 | 85.61±0.01 | 84.77±0.00 |
| halfcheetah-medium-replay | 67.38±0.34 | 55.90±0.00 | 70.58±0.84 | 69.79±0.01 |
| halfcheetah-medium | 59.88±5.40 | 52.70±0.00 | 60.87±0.89 | 60.87±0.23 |
| hopper-expert | 166.76±0.58 | 160.53±0.00 | 159.18±4.11 | 208.67±0.02 |
| hopper-medium-expert | 168.00±1.18 | 128.25±2.82 | 182.47±0.03 | 182.68±0.02 |
| hopper-medium-replay | 151.59±0.57 | 55.53±0.84 | 157.24±0.00 | 157.24±0.00 |
| hopper-medium | 143.92±1.05 | 65.94±0.26 | 136.53±0.23 | 136.53±0.20 |
| walker2d-expert | 117.25±0.46 | 113.78±0.00 | 116.64±0.14 | 116.64±0.02 |
| walker2d-medium-expert | 117.73±0.35 | 70.79±0.02 | 118.11±0.00 | 118.39±0.00 |
| walker2d-medium-replay | 92.96±0.60 | 71.83±0.05 | 93.10±0.94 | 87.35±0.07 |
| walker2d-medium | 87.65±0.40 | 66.78±0.01 | 87.11±0.07 | 87.11±0.01 |
| **Subset Avg** | 112.47±0.98 | 82.09±0.34 | 112.75±1.18 | 115.97±0.12 |
| large-diverse | 51.33±7.07 | 53.33±7.16 | 33.33±6.87 | 46.67±7.06 |
| large-play | 90.00±5.48 | 62.00±7.79 | 28.00±6.70 | 59.33±7.01 |
| medium-diverse | 93.33±4.99 | 82.67±7.98 | 58.00±7.03 | 68.00±6.83 |
| medium-play | 67.33±6.85 | 56.00±6.58 | 56.67±7.04 | 74.67±6.59 |
| **Full Avg** | 75.50±6.10 | 63.50±7.38 | 44.00±6.91 | 62.17±6.87 |

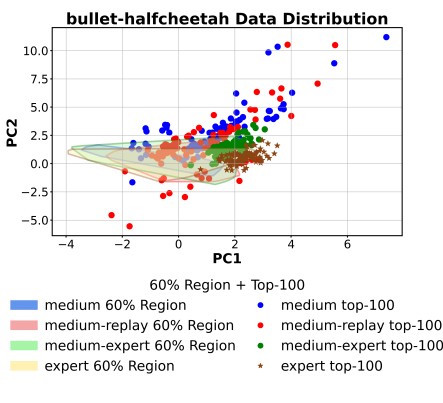

(a) Data distribution

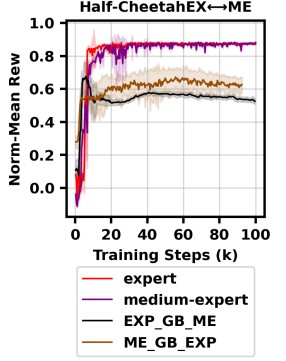

(b) Cross experiment expert & median Expert

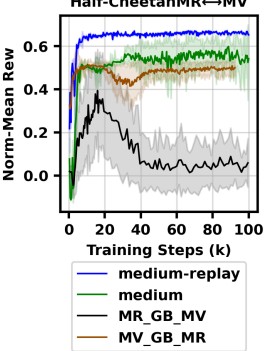

(c) Cross experiment medium & medium Replay

Figure 5: Cross-training of different data levels

Table 9: Default DQl with Different Guidance Strengths Experimental Results for Expert Adroit

| | relocate | | pen | | hammer | | door | |
|---|---|---|---|---|---|---|---|---|
| | mean | std | mean | std | mean | std | mean | std |
| $\eta=0$(BC) | 105.9296 | ±0.843 | 144.8144 | ±5.58 | 129.2588 | ±1.261 | 105.9296 | ±0.843 |
| $\eta=0.05$ | 105.4705 | ±0.906 | 141.9434 | ±6.13 | 128.5337 | ±1.273 | 105.4705 | ±0.906 |
| $\eta=0.1$ | 105.6861 | ±1.049 | 130.3630 | ±7.05 | 129.2724 | ±1.091 | 105.6861 | ±1.049 |
| $\eta=1$ | -0.2692 | ±0.2 | 36.6698 | ±7.8345 | 0.1452 | ±2.2361 | -0.1484 | ±1.4142 |

# 6 Related Work

**Modular and Decoupled Training:** Modular training has long been pursued as a desirable paradigm. In the context of diffusion models, the earliest attempts at modularity are image generation tasks that can be traced to classifier guidance (Dhariwal & Nichol, 2021), where a pretrained diffusion model was paired with a separately trained classifier to steer sampling. This approach was unstably designed to amplify the possibility of one class while suppressing all others, ignoring that features are often shared across categories, leading to distorted and fragile guidance. Recently, energy-based guidance (Lu et al., 2023) was proposed, where a diffusion model is trained first and an energy model is subsequently learned to provide guidance. However, such post-hoc modularization has proven fragile in practice—e.g., in our own experiments, more than half of the runs diverged. In contrast, our method inverts this order: we first train a guidance module using supervised learning from offline data, and then use this frozen module to learn a diffusion. It serves as a general-purpose enhancement to existing architectures of CFG. Another line of research is semi-modularized: although IDQL can be seen as a modular paradigm—learning Q-values first and then selecting one-hot action in the inference stage—it does not apply guidance during training, and its performance often lags behind joint methods such as DQL. The importance of stability under distributional shift has been repeatedly mention in classic offline RL algorithms such as BCQ (Fujimoto et al., 2019b), CQL (Kumar et al., 2020), and BRAC (Wu et al., 2019), our work proposes a guidance-first modular framework that enhances training in offline RL can also be considered as a batch-constrained or conservative regularization, using the pretrained guidance to regularize diffusion process. Finally, modular training is valid, as on vision or text diffusion-based generation widely exists; large language models and vision language models have Retrieval Augmented Generations or other special models separately trained as building blocks; however, since these modules use the web-harvested data, the limitations of OOD are not well-discussed. Offline RL must contend with severe out-of-distribution issues, making modularization brittle. This challenge has been extensively studied, with methods such as BRAC (Wu et al., 2019), CQL (Kumar et al., 2020), MOPO (Yu et al., 2020), and AWAC (Nair et al., 2020) proposing different strategies to mitigate extrapolation error. Our study provides the first systematic examination of when modular guidance is feasible and demonstrates that, under the challenges of offline RL, modular training can succeed if designed around guidance-first principles.

**Plug-and-Play Modular Composition.** The idea of plug-and-play composability is to treat pretrained modules—originally developed for other purposes—as reusable building blocks. Such composability is rarely studied and highly empirical. We propose that plug-and-play composition requires distributional alignment. For instance, in diffusion models, CLIP-based guidance (Nichol et al., 2021; Ramesh et al., 2021) applied a pretrained vision–language model as guidance, but fails on noisy intermediate states, where distribution mismatch undermines compositionality. Circumventions exist: some prior works explore modular policy composition (Andreas et al., 2017; Peng et al., 2019), focusing on skill chaining or subpolicy selection. These methods apply the plug-ins as subproblem solvers in heuristic models instead of direct reward guidance. A common challenge concerns **I/O calibration** between modules' signal magnitude. This issue is acute in systems with complex plug-in connections, such as adapter-based methods (Mou et al., 2023; Zhang et al., 2023; Ye et al., 2023), which rely heavily on the backbone's feature space. When transferred across architectures (e.g., from SD1.5 to SDXL), their performance collapses due to representational mismatch, showing these adapters are not universal interfaces. The same logic applies in NLP, where Retrieval-Augmented Generation (RAG) (Lewis et al., 2020) often suffers when the retriever is misaligned. Related retrieval-augmented frameworks such as REALM (Guu et al., 2020), FiD (Izacard & Grave, 2021), and Atlas (Izacard et al., 2022) further highlight the need for careful design of retriever–generator alignment. Overall, the design of plug-and-play models requires careful consideration to ensure alignment and stability.

Finally, plug-and-play can be beneficial if properly implemented. Value-based RL methods often suffer from intrinsic bias, since a single network trained on limited data can easily over- or under-estimate values in unseen regions. A classical remedy is *Double Q-learning* (van Hasselt, 2010; van Hasselt et al., 2016), which decouples action selection and evaluation using two networks. Our work follows this family of ideas: inspired by Double Q-learning and ensemble learning, we leverage two independently initialized modules to cancel stochastic biases inherent in a single model, a technique widely adopted in RL but underexplored in diffusion-based policies.

**Relation to Inference-Time Alignment.** Inference-time alignment for diffusion models as policies, enabling controllable generation without retraining the generator (Uehara et al., 2025). They are typically formalized using ordinary, partial, or stochastic differential equations describing diffusion sampling dynamics (Dhariwal & Nichol, 2021). Two main branches of Inference-Time Alignment models are Model-agnostic methods and Model-specific methods. Model-agnostic methods exploit diffusion stochasticity through restarts, ensemble aggregation, or noise search as inference-time compute scaling (Li et al., 2023; Ma et al., 2025). Model-specific approaches formulate alignment as sampling from reward-tilted distributions modifying the original diffusion objective (Uehara et al., 2025). Among Model-specific methods, Gradient-based methods inject reward, value, or classifier gradients into denoising dynamics, including classifier and classifier-free guidance (Dhariwal & Nichol, 2021; Chung et al., 2023). Such methods require strong alignment between guidance models and diffusion policies, and remain underexplored in offline reinforcement learning (Yang et al., 2023). Our work contributes is closest to this direction, but also has fundamental difference from it. Derivative-free approaches instead post-process generated samples using candidate selection, importance sampling, or sequential Monte Carlo resampling (Ramesh et al., 2022; Zhang et al., 2024). Bayesian SMC methods rely on Feynman–Kac formulations, emphasizing posterior correction through weighting rather than policy re-learning (Skreta et al., 2025; Doucet et al., 2009).

Our work is orthogonal to the research on inference-time alignment, which focuses on goal-oriented post-processing, such as selection or resampling, applied to a fixed, pre-trained model. Instead, we modify the training and composition of the model itself: we reorder the training pipeline via Guidance-First Diffusion Training (GFDT) to ensure guidance convergence. We also demonstrate that independently trained guidance and diffusion modules can be recombined at inference time without performance degradation. This modular training perspective reveals a form of compositional flexibility that is not captured by existing inference-time alignment frameworks, and it provides a practical pathway for reusing and recombining reinforcement learning policy components across algorithms.

## 7 Practical Considerations

In practice, we observe that during the early stage of training, the reconstruction loss overwhelmingly dominates the overall objective, effectively outnumbering the guidance loss. As a result, for roughly the first twenty thousand gradient steps, the model primarily focuses on reconstruction, and the influence of guidance remains negligible. Once the reconstruction error has sufficiently decreased, the effect of guidance becomes more visible—its gradients start shaping the diffusion behavior toward higher-reward regions.

We also notice that in some cases, both in the baselines and in GFDT, applying a guidance signal can lead to unstable training behavior, occasionally causing a failed training performance collapse after initial improvement. The inverse V shape curve in Fig.5 is a good example when the guidance is not applied properly. However, OOD guidance issue is not intrinsic, but occasional failing can be fixed. A simple and effective remedy is to gradually introduce the guidance term—for example, linearly increasing its weight after the first few thousand training steps and finally applying the full weight to the model—which empirically stabilizes training and consistently prevents such failures. In our experiments, after applying this trick, training failure has never occurred.

## Reproducibility and Ethics Statement

Experiments use three random seeds; hyperparameters are in Appendix C. Code, pretrained models, and scripts are available at `https://github.com/modulardiffusion-design/Modular_diffusion`. The study uses public benchmarks only and has no foreseeable negative societal impact, though safe and fair deployment should be considered.

# A   Algorithmic Details

## A.1   GFDT Algorithmic Details

The entire training and inference process of GFDT is shown in Fig.6, as well as the algorithm1.

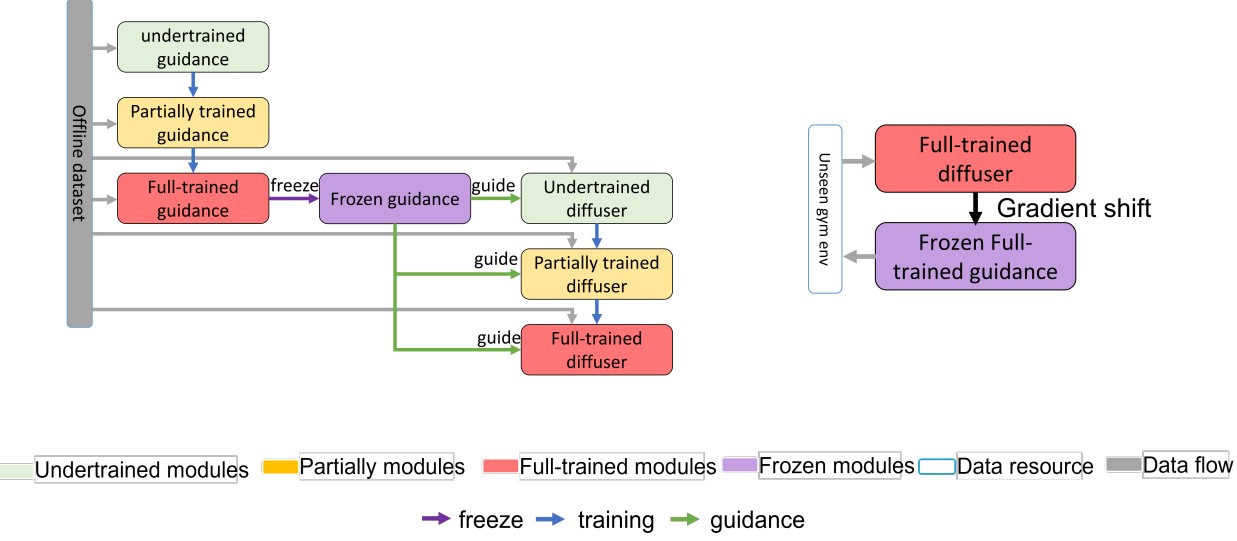

Figure 6: Training and inference stage of GFDT

**Important explanation of the frozen Q network still updating the diffusion network** The details of GFDT have been thoroughly explained in the main content. A reasonable concern is the role of $Q_\phi$. Although $Q_\phi$ is frozen and does not update the parameters of the guidance network, its output values are still used to adjust the parameters of the diffusion network, encouraging it to generate samples with higher Q values.

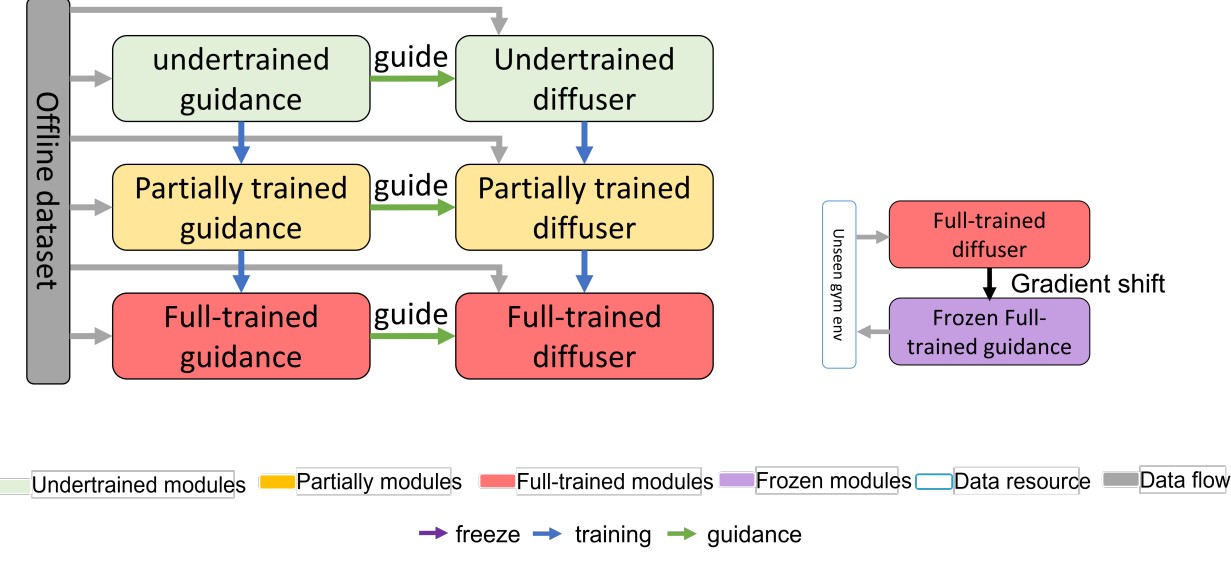

Figure 7: Training and inference stage of traditional CFG

---

**Algorithm 2** Traditional Gradient Guidance Training

---

**Require:** Offline dataset $\mathcal{D} = \{(s, a, r, s')\}$, diffusion model $\epsilon_\theta$, Q-function $Q_\phi$ (learned jointly), training steps $N_\theta$
1: **for** $j = 1$ to $N_\theta$ **do**
2:      Sample $(s, a_0, r, s') \sim \mathcal{D}$
3:      TD target: $y \leftarrow r + \gamma \cdot \max_{a'} Q_\phi(s', a')$
4:      Update $Q_\phi$ to minimize $\mathcal{L}_Q = \|Q_\phi(s, a_0) - y\|^2$
5:      Add noise: $a_t \leftarrow \sqrt{\bar{\alpha}_t}\, a_0 + \sqrt{1 - \bar{\alpha}_t} \cdot \epsilon, \quad \epsilon \sim \mathcal{N}(0, I)$
6:      Predict noise: $\hat{\epsilon} \leftarrow \epsilon_\theta(a_t, s, t)$
7:      Update $\epsilon_\theta$ to minimize $\mathcal{L}_{\text{diff}} = \|\hat{\epsilon} - \epsilon\|^2 + \mathcal{L}_Q$
8: **Return** trained $\epsilon_\theta$, jointly-trained $Q_\phi$
9:
10: **Inference:**
11: **for** denoise steps **do**
12:      Sample candidate $a_k \sim \pi_\theta(\cdot|s)$
13:      Apply guidance: $a_{k-1} \leftarrow a_k + \lambda \cdot \nabla_a Q_\phi(s, a_k)$
14: **Return** $a^\star$

---

## A.2 IDQL Algorithmic Details

**Training.** Implicit Diffusion Q-learning (IDQL) decouples the training of the diffusion model from the Q-value estimator. The diffusion policy $\pi_\theta(a|s)$ is trained purely by behavior cloning (BC) from the offline dataset $\mathcal{D} = \{(s, a)\}$, i.e.,

$$\min_\theta \; \mathbb{E}_{(s,a) \sim \mathcal{D}} \left[ \|a - \pi_\theta(s)\|^2 \right], \tag{11}$$

without any reward or Q-guidance incorporated into the diffusion process. In parallel, a separate Q-network $Q_\phi(s, a)$ is learned by standard temporal-difference (TD) regression:

$$\min_\phi \; \mathbb{E}_{(s,a,r,s') \sim \mathcal{D}} \left[ \left( Q_\phi(s, a) - (r + \gamma \max_{a'} Q_\phi(s', a')) \right)^2 \right]. \tag{12}$$

Notably, the Q-network is updated together with the diffusion model during training, although it does not directly affect the diffusion optimization.

**Inference.** At deployment, the diffusion model generates $n$ candidate actions $\{a_1, a_2, \ldots, a_n\}$ sampled from the diffusion model $\pi_\theta(\cdot|s)$. These are then evaluated with the learned Q-network, and the action with the highest Q-value is selected:

$$a^\star = \arg \max_{i=1,\ldots,K} Q_\phi(s, a_i). \tag{13}$$

This procedure can also be interpreted as a one-hot selection mechanism over the candidate actions, where the Q-network acts as a ranking function. The algorithm of IDQL is in

**Comparison between GFDT and IDQL** The key distinction between IDQL and our proposed GFDT lies in how Q-guidance is incorporated. In IDQL, a separate Q-guidance network is trained, but reward information is not injected into the diffusion model during training. This design makes IDQL relatively stable under the distribution, since the learned policy is not directly biased by Q-values that could otherwise push the distribution toward out-of-distribution regions. However, the downside is that the policy is less reward-optimal, because it is guided primarily by behavioral cloning rather than directly exploiting reward signals. At inference time, IDQL applies reward guidance only as a one-hot selection among the generated actions. While this procedure ensures the selected actions are rewarding, they remain restricted to the support of the behaviorally cloned generator, which limits the achievable performance compared to gradient-guided methods.

In contrast, GFDT explicitly integrates Q-guidance into the generative diffusion process. Reward information actively shapes the learned distribution during training, which leads to more reward-aligned behaviors. At inference time, GFDT further applies a gradient shift, using gradient descent to nudge the generated

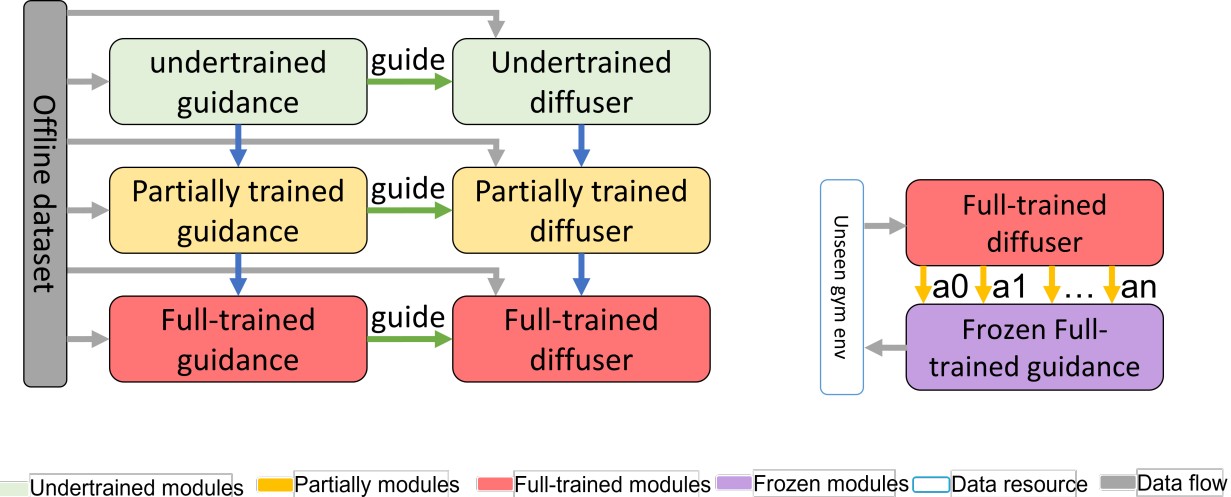

Figure 8: Training and inference stage of traditional CFG.

---

**Algorithm 3** Implicit Diffusion Q-learning (IDQL)

---

1: **Input:** offline dataset $\mathcal{D}$
2: Initialize diffusion policy $\pi_\theta$ and Q-network $Q_\phi$
3: **while** not converged **do**
4:     Sample batch $(s, a, r, s')$ from $\mathcal{D}$
5:     Update $\pi_\theta$ by behavior cloning on $(s, a)$
6:     Update $Q_\phi$ by TD regression on $(s, a, r, s')$

7: **Inference:**
8: **Input:** state $s$
9: **for** $k = K, K-1, \ldots, 1$ **do**
10:     **for** $n = 1, \ldots, N$ **do**
11:         Sample candidate action $a_n^{(k)} \sim \pi_\theta(\cdot \mid s)$
12:         Evaluate $Q_\phi(s, a_n^{(k)})$
13:     Select best action at step $k$:
$$a^{(k)\star} = \arg \max_{n=1,\ldots,N} Q_\phi(s, a_n^{(k)})$$

14: **Return** $a^{(1)\star}$

---

action toward the locally optimal point. As a result, GFDT consistently outperforms both the baseline CFG and IDQL in experiments. Although gradient-based guidance has the potential to destabilize training, normalization and a staged training scheme mitigate this risk: reconstruction dominates in the early phase and Q-guidance gradually takes effect afterwards, and therefore, destabilized training can manifest a certain V-shape mode. (see Appendix G).

## B  Trajectory-based Guidance: Why it Fails in Offline RL

This algorithm currently does not apply to trajectory-based methods. We trained several trajectory-return guided variants; however, the guidance estimates did not *converge* in the actionable sense: they moved from near-zero to a coarse range (e.g., $200-300$ in a median expert dataset with Q values of $\approx 200-300$) but could not refine within that band, making the guidance ineffective. If the guidance itself is not converged, our method also does not make a difference. We attribute this to (i) noisy returns and (ii) long-horizon credit

assignment, both exacerbated in offline settings without on-policy rollouts. By contrast, TD-based $(s, a)$-level guidance provides stable, local gradients that are compatible with diffusion updates.

## C    Experimental Setup and Hyperparameters

.All models are trained using the D4RLMuJoCoTD Dataset (Fu et al., 2021). It provides pre-collected trajectories of varying quality, including expert, medium-expert, medium-replay, and medium datasets, enabling rigorous training of offline RL algorithms under diverse data distributions. The evaluation is done in randomly initialized environments. All the gradient steps mentioned are with respect to a batch size of 256.

Wherever possible, we adopt the original hyperparameter settings from the paper of (Wang et al., 2024). Intentionally, we do not modify any training-related components—including the optimizer, learning rate, batch size, architecture, or loss function. Because a key strength of our method is that it achieves superior performance without requiring any changes to the other parameters other than pretraining or modularize. This highlights the robustness and plug-and-play nature of our approach. Final results are reported, averaged over 50 evaluation episodes. Performance is measured by normalized return, and we report both the mean(Section 4.1) ).

All experiments were implemented in PyTorch and based on the CLEANDIFFUSER framework (Wang et al., 2024). We strictly followed the hyperparameter settings from the baseline implementations to ensure fair comparison. Table 10 summarizes the key values.

Table 10: Hyperparameters used in our experiments (inherited from CleanDiffuser).

| Parameter | Value | Notes |
|---|---|---|
| Optimizer | Adam | |
| Learning Rate | $3 \times 10^{-4}$ | fixed across all models |
| Batch Size | 256 | |
| Discount Factor $\gamma$ | 0.99 | |
| Noise Schedule | cosine | unless otherwise specified |
| Number of Diffusion Steps $T$ | 5,10,20,30,40 | |
| Actor Loss Weight $\eta$ | 1.0 | scales $\mathcal{L}_Q$ |

For reproducibility, we will release full training scripts and environment configurations in our code repository.

## D    Why the generated action with the GFDT and the modular methods are in distribution

**Addressing a Key Concern**    One might naturally worry that even if the guidance module (e.g., the Q-function $Q_\phi$) and the diffusion model are both trained solely on the offline dataset $\mathcal{B}$, their combination during sampling could still produce out-of-distribution actions. The value guidance term $\nabla_a Q_\phi(a)$ may have a large magnitude or steep gradients, which could push the sampled action $a_t$ away from the data manifold if not properly controlled. Therefore, the perturbation magnitude and the guidance magnitude are tightly controlled, and the guidance gradient is normalized :

$$\|a_{t+1} - a_t\| \leq \lambda + \mathcal{O}(\sqrt{\tau_t}), \lambda > 0 \ and \ \lambda \to 0$$

ensuring that each sampling step stays within a small neighborhood of the current point. Since the diffusion model is trained on the dataset $\mathcal{B}$ and generates samples close to it, and since the guidance is applied as a *soft* correction, the overall sampling trajectory remains near the support of $\mathcal{B}$. Thus, even when guided by $Q_\phi$, the diffusion process remains effectively batch-constrained.

## E    Comparison Table of GFDT

This section contains two performance comparison tables Table.11 and Table. 12, that shows the percentages of improvement of GDFT and as other methods, compared to baseline models.

Table 11: Mujoco and Antmaze results of DQL. Each cell shows the raw score and the relative performance (%).

| Environment | Baseline | GFDT | GFDT(%) | GAI | GAI(%) | BC | BC(%) | Unfreeze(%) |
|---|---|---|---|---|---|---|---|---|
| halfcheetah-expert | 88.21 | 90.43 | 102.51% | 68.49 | 77.65% | 85.82 | 97.29% | 101.47% |
| halfcheetah-medium-expert | 88.28 | 90.18 | 102.15% | 89.73 | 101.64% | 85.53 | 96.88% | 101.67% |
| halfcheetah-medium-replay | 67.38 | 67.93 | 100.81% | 67.76 | 100.57% | 55.87 | 82.92% | 100.08% |
| halfcheetah-medium | 59.88 | 66.99 | 111.88% | 53.36 | 89.12% | 54.57 | 91.13% | 91.93% |
| hopper-expert | 166.76 | 172.90 | 103.68% | 162.68 | 97.55% | 165.11 | 99.01% | 93.03% |
| hopper-medium-expert | 168.00 | 172.31 | 102.56% | 166.28 | 98.98% | 166.88 | 99.33% | 94.70% |
| hopper-medium-replay | 151.59 | 152.98 | 100.91% | 121.97 | 80.46% | 113.30 | 74.74% | 99.28% |
| hopper-medium | 143.92 | 147.06 | 102.18% | 71.09 | 49.40% | 118.08 | 82.04% | 100.47% |
| walker2d-expert | 117.25 | 120.25 | 102.56% | 119.91 | 102.28% | 108.05 | 92.16% | 100.57% |
| walker2d-medium-expert | 117.73 | 117.63 | 99.91% | 115.08 | 97.75% | 117.45 | 99.76% | 100.65% |
| walker2d-medium-replay | 92.96 | 95.62 | 102.86% | 80.32 | 86.40% | 87.14 | 93.74% | 102.86% |
| walker2d-medium | 87.65 | 87.89 | 100.27% | 77.33 | 88.22% | 82.28 | 93.87% | 100.27% |
| Average | 112.47 | 115.18 | 102.69% | 99.50 | 89.17% | 103.34 | 91.91% | 98.91% |
| antmaze-large-diverse-v2 | 63.33 | 90.67 | 143.16% | 86.67 | 136.84% | 66.00 | 104.21% | 136.84% |
| antmaze-large-play-v2 | 90.00 | 89.33 | 99.26% | 88.67 | 98.52% | 50.67 | 56.30% | 101.48% |
| antmaze-medium-diverse-v2 | 93.33 | 97.33 | 104.29% | 94.00 | 100.71% | 78.00 | 83.57% | 102.86% |
| antmaze-medium-play-v2 | 67.33 | 91.33 | 135.64% | 88.67 | 131.68% | 68.00 | 100.99% | 138.61% |
| Average | 78.50 | 92.17 | 120.59% | 89.50 | 116.94% | 65.67 | 86.27% | 119.95% |

Table 12: Mujoco and Antmaze results of EDP. Each cell shows the raw score and the relative performance (%).

| Environment | EDP_baseline | GFDT | GFDT(%) | GAI | GAI(%) | BC | BC(%) | Unfreeze(%) |
|---|---|---|---|---|---|---|---|---|
| halfcheetah-expert | 86.55 | 86.82 | 100.30% | 78.78 | 91.02% | 85.82 | 99.15% | 99.34% |
| halfcheetah-medium-expert | 86.74 | 87.16 | 100.48% | 78.18 | 90.14% | 85.53 | 98.61% | 100.12% |
| halfcheetah-medium-replay | 65.78 | 64.42 | 97.94% | 47.08 | 71.57% | 55.87 | 84.94% | 97.94% |
| halfcheetah-medium | 54.59 | 59.26 | 108.57% | 53.56 | 98.12% | 54.57 | 99.96% | 101.36% |
| hopper-expert | 161.23 | 163.65 | 101.50% | 151.93 | 94.23% | 165.11 | 102.41% | 100.63% |
| hopper-medium-expert | 161.36 | 163.95 | 101.61% | 141.07 | 87.43% | 166.88 | 103.42% | 101.44% |
| hopper-medium-replay | 112.16 | 139.75 | 124.60% | 114.32 | 101.93% | 113.30 | 101.02% | 106.91% |
| hopper-medium | 141.16 | 141.64 | 100.35% | 83.25 | 58.98% | 118.08 | 83.65% | 100.27% |
| walker2d-expert | 116.14 | 116.34 | 100.18% | 114.27 | 98.39% | 108.05 | 93.04% | 100.19% |
| walker2d-medium-expert | 116.31 | 116.59 | 100.24% | 109.55 | 94.19% | 117.45 | 100.98% | 100.65% |
| walker2d-medium-replay | 85.02 | 83.77 | 98.53% | 68.84 | 80.97% | 87.14 | 102.49% | 100.19% |
| walker2d-medium | 84.24 | 84.21 | 99.96% | 58.21 | 69.10% | 82.28 | 97.67% | 100.24% |
| Average | 105.94 | 108.96 | 102.86% | 91.59 | 86.34% | 103.34 | 97.28% | 100.77% |
| antmaze-large-diverse-v2 | 30.67 | 34.67 | 113.05% | 28.67 | 93.48% | 10.67 | 34.78% | 154.35% |
| antmaze-large-play-v2 | 21.33 | 22.67 | 106.25% | 18.67 | 87.50% | 18.67 | 87.50% | 118.75% |
| antmaze-medium-diverse-v2 | 67.33 | 52.00 | 77.23% | 2.00 | 2.97% | 18.00 | 26.73% | 19.80% |
| antmaze-medium-play-v2 | 73.30 | 118.00 | 160.98% | 90.00 | 122.78% | 76.00 | 103.68% | 154.62% |
| Average | 48.16 | 56.83 | 114.38% | 34.83 | 76.68% | 30.83 | 63.17% | 111.88% |

# F    Detailed Analysis of Plug-and-play

Table 13: Training gradient steps to get 95% performance and parameter statistics summary (batch size 256)

| Env | DQL | GFDT_DQL | EDP | GFDT_EDP | DDIG | IDDG |
|---|---|---|---|---|---|---|
| HCEX | 7600 | 2800 (36.84%) | 18000 | 3600 (20.00%) | 13200 (63.46%) | 10400 (50.00%) |
| HCME | 16400 | 2800 (17.07%) | 31200 | 6000 (19.23%) | 26400 (81.48%) | 14400 (44.44%) |
| HCMR | 8000 | 3200 (40.00%) | 20400 | 8400 (41.18%) | 7200 (26.47%) | 35600 (130.90%) |
| HCM | 52000 | 7600 (14.62%) | 10800 | 10000 (92.59%) | 6400 (7.24%) | 45600 (51.60%) |
| HOEX | 30800 | 15200 (49.35%) | 8400 | 8400 (100.00%) | 38400 (118.52%) | 4400 (13.60%) |
| HOME | 41600 | 23600 (56.73%) | 24000 | 16800 (70.00%) | 44400 (105.71%) | 56000 (133.30%) |
| HOMR | 24800 | 3200 (12.90%) | 40800 | 52800 (129.41%) | 71200 (127.14%) | 35600 (63.60%) |
| HOM | 79200 | 12800 (16.16%) | 50400 | 76800 (152.38%) | 90800 (76.95%) | 19200 (16.30%) |
| WAEX | 59600 | 14000 (23.49%) | 10800 | 7200 (66.67%) | 51200 (69.57%) | 30400 (41.30%) |
| WAME | 96000 | 72400 (75.42%) | 21600 | 18000 (83.33%) | 47600 (42.65%) | 31600 (28.30%) |
| WAMR | 26000 | 9600 (36.92%) | 38400 | 10800 (28.13%) | 10800 (16.07%) | 400 (0.60%) |
| WAM | 10800 | 3600 (33.33%) | 79200 | 51600 (65.15%) | 24000 (55.05%) | 60400 (138.50%) |
| AVG | 37733 | 14233 (34.40%) | 29500 | 22533 (72.34%) | 35967 (65.86%) | 28667 (59.37%) |
| Ldiv | 500000 | 47600 (9.52%) | 1600000 | 900000 (56.25%) | 64800 (12.96%) | 400000 (80.00%) |
| Lplay | 1300000 | 29200 (2.25%) | 1300000 | 87600 (6.74%) | 76400 (5.88%) | 70800 (5.45%) |
| Mdiv | 1500000 | 18000 (1.20%) | 800000 | 17200 (2.15%) | 37600 (2.51%) | 44400 (2.96%) |
| Mplay | 89600 | 15200 (16.96%) | 1400000 | 1400000 (100.00%) | 85200 (95.09%) | 43200 (48.21%) |
| AVG | 847400 | 27500 (7.48%) | 1275000 | 601200 (41.28%) | 66000 (29.11%) | 139600 (34.16%) |

Table 14: Ablation study on slow guidance

| Env | EDP | | | | DQL | | | |
|---|---|---|---|---|---|---|---|---|
| | baseline | GFDT | 100 | slow_guid | baseline | GFDT | 100 | show_guid |
| HCEX | 86.55 | 86.82 | 81.3680 | 73.5786 | 88.21 | 90.43 | 77.9011 | 75.3834 |
| HCME | 86.74 | 87.16 | 81.0966 | 74.9345 | 88.28 | 90.18 | 79.5332 | 73.9830 |
| HCMR | 65.78 | 64.42 | 63.7889 | 52.7860 | 67.38 | 67.93 | 64.3091 | 42.6733 |
| HCMV | 54.59 | 59.26 | 53.1212 | 52.3981 | 59.88 | 66.99 | 51.9504 | 51.5571 |
| HOEX | 161.23 | 163.65 | 162.5470 | 156.0822 | 166.76 | 172.90 | 169.2275 | 125.8843 |
| HOME | 161.36 | 163.95 | 158.4901 | 144.5272 | 168.00 | 172.31 | 158.5458 | 47.8485 |
| HOMR | 112.16 | 139.75 | 54.0879 | 40.7498 | 151.59 | 152.98 | 149.9167 | 58.9052 |
| HOMV | 141.16 | 141.64 | 80.5823 | 82.1743 | 143.92 | 147.06 | 136.6000 | 81.3338 |
| WAEX | 116.14 | 116.34 | 115.3033 | 113.8683 | 117.25 | 120.25 | 115.0718 | 113.2748 |
| WAME | 116.31 | 116.59 | 105.3806 | 94.4389 | 117.73 | 117.63 | 89.3075 | 92.1322 |
| WAMR | 85.02 | 83.77 | 80.3006 | 35.2995 | 92.96 | 95.62 | 85.0038 | 10.6141 |
| WAMV | 84.24 | 84.21 | 70.9460 | 52.2359 | 87.65 | 87.89 | 68.8932 | 66.3106 |
| avg | 105.94 | 108.96 | 92.2510 | 81.0894 | 112.47 | 115.18 | 103.8550 | 69.9917 |

## G   Details of Application Scope

These are all the plots of cross experiments. We did not perform data mixing experiments because, as shown in the study by Miao et al. (2023), mixing datasets Generated from different policies can lead to degraded performance. Our ablation on the OOD dataset does not involve dataset mixing, since the diffusion model and the guidance model are each trained on only one dataset. Consequently, the diffusion model does not learn from multiple policies. However, when the guidance model is trained on a different dataset, it may fail to provide correct gradient signals.

Table 15: Ablation study on half trained guidance

| Env | EDP | | | | DQL | | | |
|---|---|---|---|---|---|---|---|---|
| | baseline | GFDT | half-trained | RAN | baseline | GFDT | half-trained | RAN |
| HCEX | 86.55 | 86.82 | 81.7248 | 21.0728 | 88.21 | 90.43 | 49.1359 | 37.9408 |
| HCME | 86.74 | 87.16 | 75.9758 | 38.0637 | 88.28 | 90.18 | 32.7013 | 8.3004 |
| HCMR | 65.78 | 64.42 | 51.1627 | 9.8432 | 67.38 | 67.93 | 34.0113 | 3.5598 |
| HCMV | 54.59 | 59.26 | 51.9935 | 42.2812 | 59.88 | 66.99 | 35.8784 | 15.2016 |
| HOEX | 161.23 | 163.65 | 159.7950 | 21.3194 | 166.76 | 172.90 | 52.1442 | 4.3527 |
| HOME | 161.36 | 163.95 | 137.2213 | 61.1886 | 168.00 | 172.31 | 163.8437 | 1.2043 |
| HOMR | 112.16 | 139.75 | 10.1941 | 31.0669 | 151.59 | 152.98 | 150.7078 | 2.8809 |
| HOMV | 141.16 | 141.64 | 71.8405 | 23.2034 | 143.92 | 147.06 | 24.5012 | 1.4580 |
| WAEX | 116.14 | 116.34 | 100.6322 | 13.2890 | 117.25 | 120.25 | -0.0793 | 56.2783 |
| WAME | 116.31 | 116.59 | 57.0844 | 80.4424 | 117.73 | 117.63 | -0.3491 | 40.4744 |
| WAMR | 85.02 | 83.77 | 25.9705 | 38.6119 | 92.96 | 95.62 | -0.2370 | 15.3321 |
| WAMV | 84.24 | 84.21 | 55.5008 | 6.5563 | 87.65 | 87.89 | -0.2266 | 23.8723 |
| avg | 105.94 | 108.96 | 73.2580 | 32.2449 | 112.47 | 115.18 | 45.1693 | 17.5713 |

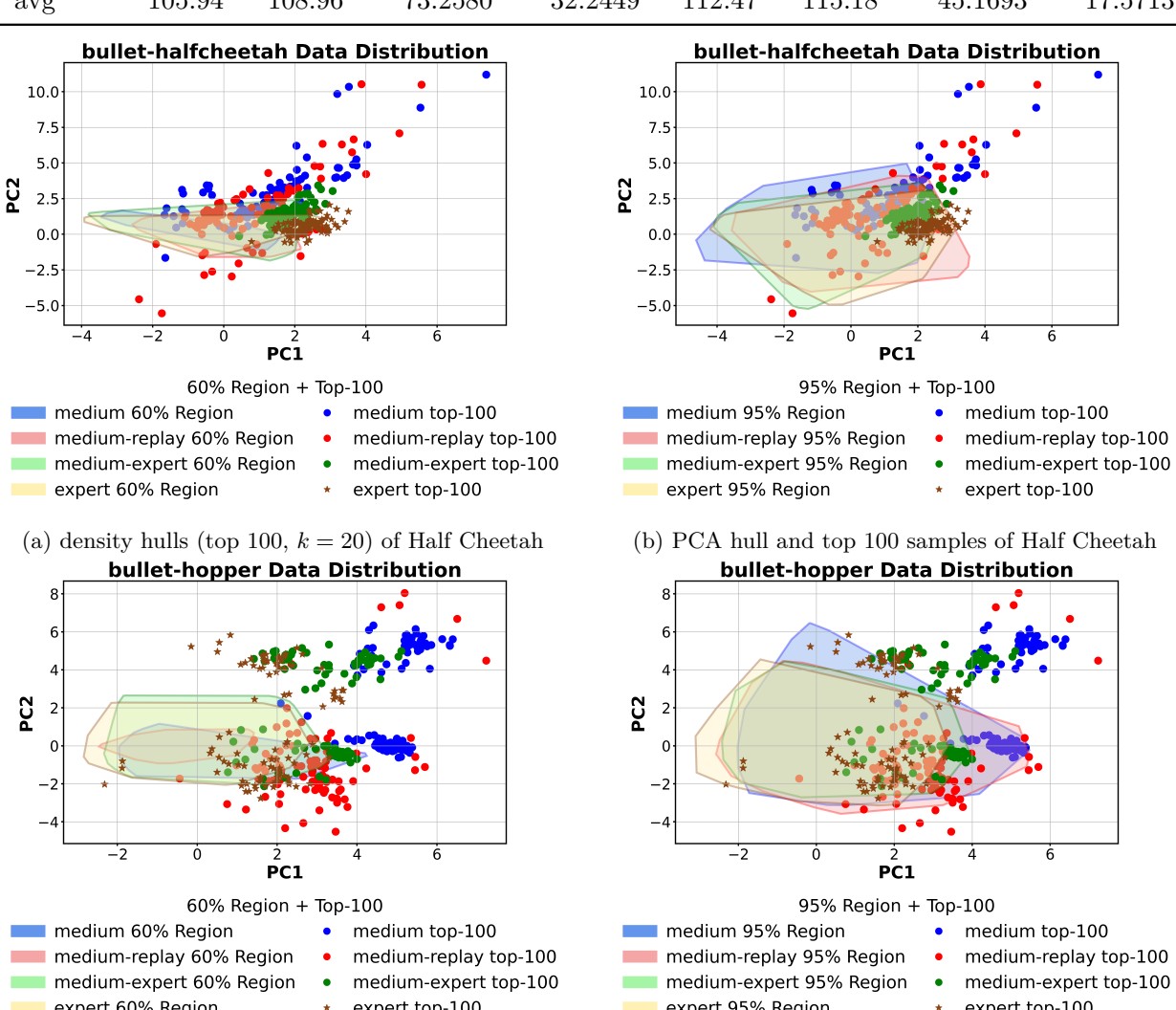

(a) density hulls (top 100, $k = 20$) of Half Cheetah    (b) PCA hull and top 100 samples of Half Cheetah

(a) density hulls (top 100, $k = 20$) of Hopper    (b) PCA hull and top 100 samples of hopper

| | DGID: DGID (DQL-Guidance + IDQL-Diffusion): DGID follows the overall pipeline of IDQL, meaning that its generation and diffusion process are still driven by IDQL's methodology. The key difference is that the component responsible for selecting optimal action candidates has been replaced by a DQL module. | IGDD: IGDD (IDQL-Guidance + DQL-Diffusion): IGDD is built upon DQL's architecture and training logic, but replaces its gradient-based optimization module with the advantage-guided update mechanism from IDQL. |
|---|---|---|
| **MUJOCO Environment Characteristics:** MuJoCo's dense-reward nature provides immediate evaluative feedback at every timestep. This continuous supervision allows the agent to quickly identify and correct suboptimal actions, effectively smoothing the learning curve. Consequently, the environment is forgiving to small deviations or approximation errors, as incorrect behaviors are rapidly penalized and adjusted. | Compared to the original IDQL, DGID achieves a significant performance improvement, with an average score of 112.75, and becomes competitive with the baseline DQL model. The DQL module offers a strong optimization signal that effectively give IDQL a most deterministically optimal guidance on the action choice, during inference. Since the MuJoCo environment provides dense and continuous rewards, the system can directly benefit from DQL's precise action evaluation. The dense-reward setting also makes the model more tolerant to small distribution mismatches between DQL and IDQL(errors corrected immediately), leading to stable and superior overall performance. | This effectively smooths the optimization trajectory and regularizes the learning process. IGDD performs slightly better than the original DQL, achieving an average score of 115.97 and showing improved stability during diffusion. The IDQL-style advantage guidance provides smoother gradients and prevents overly aggressive Q-value maximization, which is a known issue in pure DQL setups. The result is a more balanced and robust optimization process that preserves DQL's iterative refinement strength while improving convergence reliability. In dense-reward environments like MuJoCo, where feedback is immediate and continuous, this regularized update leads to steady and consistent performance gains. |
| **antmaze:** the task follows a typical sparse-reward setting, where the agent only receives a positive signal upon successful completion of the maze. Due to the extremely long horizons—often spanning thousands of steps—the model requires high precision and the accumulated errors are not easily corrected. Moreover, the environment is heavily affected by noise and stochastic dynamics, making it highly sensitive to approximation errors. | It is suspected that When the DQL guidance is applied to out-of-distribution actions generated by IDQL, which does not generate the most rewarding signal every step, its Q-values become unreliable and often misleadingly high.(recall that mujoco is less noisy and easy to correct errors). The sharp, deterministic MaxQ operator further amplifies these estimation errors, leading to unstable and degraded performance. The main reason for the lower performance of PAP is antmaze needs conservative high precision and the broken of assumption of "small enough steps and long enough trajectory" became several in this sensitive environment. | (DGID follows IDQL's pipeline, but replaces its action selection with DQL's MaxQ-guided module). IGDD achieves a suboptimal result in AntMaze, with an average score of 62.17. It underperforms relative to original DQL but remains noticeably better than DGID. Because the IDQL-style guidance provides weaker but smoother signals, it is insufficient for the sparse-reward nature of AntMaze. This environment only provides a final success reward, so precise long-horizon navigation depends on sharp gradient signals to correctly guide early steps. The lack of strong feedback causes gradual error accumulation across iterations, leading to less efficient exploration and reduced success rates. |

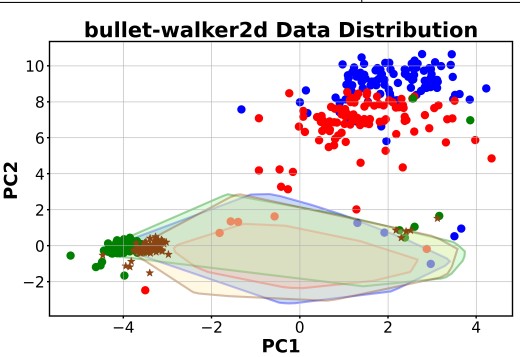

**bullet-walker2d Data Distribution**

60% Region + Top-100

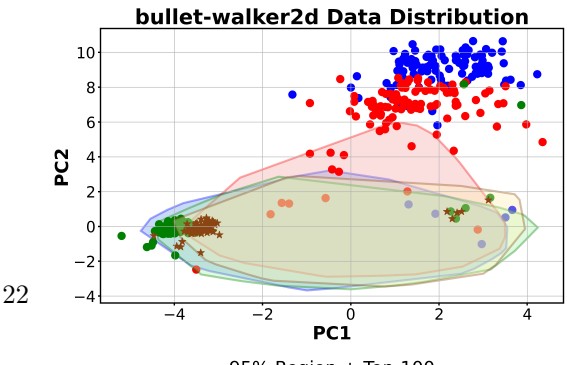

**bullet-walker2d Data Distribution**

95% Region + Top-100

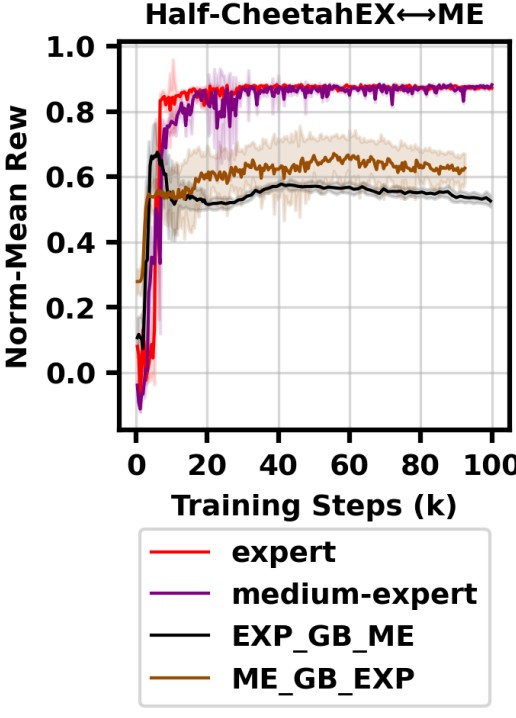

(a) Across Experiments of Expert (EX) and Medium Expert (ME)

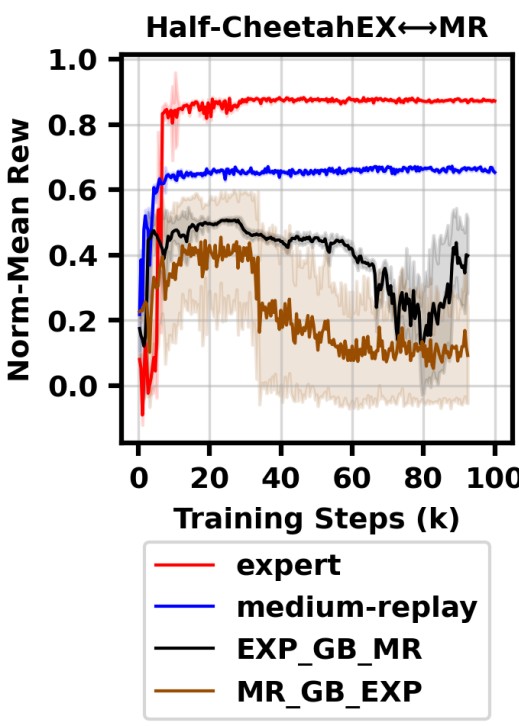

(b) Across Experiments of Expert(EX) and Medium Replay(MR)

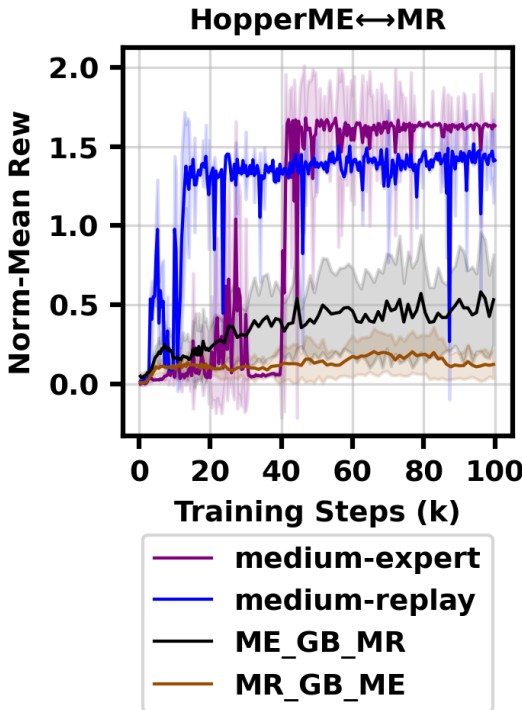

(c) Across Experiments of Medium Expert(ME) and Medium Replay(MR)

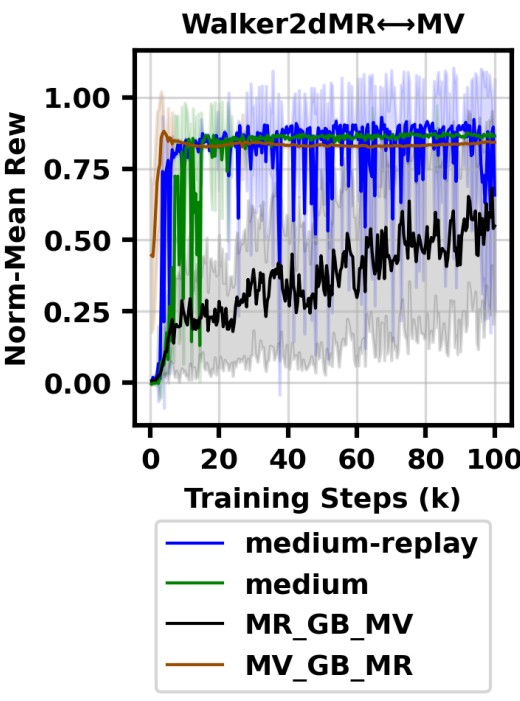

(d) Across Experiments of Medium Replay(MR) and Medium(MV)

## H    Notation and Terminology

| Symbol | Meaning |
| --- | --- |
| $\mathcal{D}$ | Offline dataset consisting of tuples $(s, a, r, s')$, collected by an unknown behavior policy and fixed throughout training. |
| $s, a, r, s'$ | State, action, reward, and next state sampled from the offline dataset $\mathcal{D}$. |
| $\pi_\theta$ | Diffusion-based policy parameterized by $\theta$. The policy is defined implicitly by a conditional reverse diffusion process, not by an explicit density. |
| $a_0$ | Clean (final) action generated by the diffusion policy or taken from the dataset. |
| $a_t$ | Noisy action at diffusion timestep $t$. |
| $\epsilon_\theta$ | Denoising network used in the diffusion policy, trained to predict the injected noise during diffusion. |
| $Q(s, a)$ | state–action value function, defined as the expected discounted return starting from $(s, a)$. |
| $Q_\phi$ | General representation. Parameterized guidance module (Q-network) trained on the offline dataset using temporal-difference learning. It serves as a different reward estimator from and provides gradient-based guidance to the diffusion model. |
| $Q_p$ | A **pretrained and frozen** guidance module. This Q-network is trained independently (e.g., via DQL or IDQL) and reused to guide diffusion training or inference without further updates. |

Table 16: Symbol definitions and meanings

Table 17: Algorithm Abbreviations

| Abbreviation | Explanation |
| --- | --- |
| Env | **Environment.** The benchmark task or environment in which the algorithm is evaluated (e.g., HalfCheetah, Hopper, Walker2d, AntMaze). |
| ReBR | **Regularized Behavior Regularized Actor Critic (ReBRAC).** A conservative offline RL algorithm emphasizing stability through behavior regularization (Wu et al., 2019). |
| DICE | **Dynamic Importance Sampling Correction Estimator.** An offline RL method that uses importance weighting to correct distribution mismatch during policy evaluation (Ma et al., 2024). |
| CQL | **Conservative Q-Learning.** A value-based offline RL algorithm that penalizes overestimation of unseen actions to ensure conservative value estimation (Kumar et al., 2020). |
| IQL | **Implicit Q-Learning.** A decoupled offline RL method that learns Q-values and implicitly defines a policy via advantage-weighted regression without explicit policy optimization (Kostrikov et al., 2022). |
| DQL_GF | **Diffusion Q-Learning with Guidance-First (GFDT).** The DQL algorithm trained under the Guidance-First Diffusion Training paradigm, where the Q-network is pretrained and frozen before diffusion training. |
| EDP_GF | **Efficient Diffusion Policy with Guidance-First (GFDT).** A one-step denoising diffusion policy algorithm enhanced with pretrained frozen guidance to accelerate convergence and improve efficiency. |
| GFDT | **Guidance-First Diffusion Training.** A training paradigm where the guidance (Q-network) is pretrained and frozen before training the diffusion policy. |
| Baseline | The original implementation of the corresponding diffusion-based offline RL algorithm (e.g., DQL, IDQL, or EDP) without our proposed modifications. |
| D-Baseline | **Double-Guidance Baseline.** A variant of the baseline model where the guidance module used during inference is replaced with an independently initialized version of the same architecture (different random seed). |
| Double_GFDT | GFDT equipped with *Double Guidance* at inference time, reducing variance through independent initialization. |
| GAI | **Guidance at Inference.** A behavior cloning model trained without reward guidance during training, but augmented with Q-guidance only at inference. |
| Unfreeze | A variant of GFDT where the pretrained guidance module is not frozen but continues to be updated during policy training. |
| BC | **Behavior Cloning.** A supervised learning baseline that trains a policy to imitate dataset actions without reward guidance. |
| DGID | **DQL-Guidance with IDQL-Diffusion.** A plug-and-play configuration where the **guidance module** is taken from DQL (D), while the **diffusion policy** is taken from IDQL (I). Here, D denotes DQL, I denotes IDQL, G denotes the guidance module, and the final D denotes the diffusion model. |
| IGDD | **IDQL-Guidance with DQL-Diffusion.** A plug-and-play configuration where the **guidance module** is taken from IDQL (I), while the **diffusion policy** is taken from DQL (D). The notation follows the same convention: I/D indicate the algorithm source (IDQL or DQL), G denotes the guidance module, and D denotes the diffusion model. |

Table 18: D4RL Benchmark Environment Abbreviations

| Abbreviation | Explanation |
|---|---|
| HCEX | **HalfCheetah-Expert.** Dataset generated by an expert policy in the HalfCheetah environment. |
| HCME | **HalfCheetah-Medium-Expert.** Dataset generated by a mixture of medium and expert policies in HalfCheetah. |
| HCMR | **HalfCheetah-Medium-Replay.** Dataset generated by replay buffer data collected from medium-performance policies. |
| HCMV | **HalfCheetah-Medium.** Dataset generated by a medium-performance policy in HalfCheetah. |
| HOEX | **Hopper-Expert.** Dataset generated by an expert policy in the Hopper environment. |
| HOME | **Hopper-Medium-Expert.** Dataset generated by a mixture of medium and expert policies in Hopper. |
| HOMR | **Hopper-Medium-Replay.** Replay buffer dataset in Hopper. |
| HOMV | **Hopper-Medium.** Dataset generated by a medium policy in Hopper. |
| WAEX | **Walker2d-Expert.** Expert policy dataset in Walker2d. |
| WAME | **Walker2d-Medium-Expert.** Mixed dataset in Walker2d. |
| WAMR | **Walker2d-Medium-Replay.** Replay dataset in Walker2d. |
| WAMV | **Walker2d-Medium.** Medium policy dataset in Walker2d. |

# I  Error bound calculation with respect to gradient steps

In this appendix, we provide a simple scaling calculation to clarify how the number of training steps depends on the desired relative error reduction. This calculation is intended purely for intuition and does not constitute a formal convergence guarantee.

We start from a generic estimation bound of the form

$$\mathbb{E}\big[\|\nabla_a Q_\phi - \nabla_a Q^*\|\big] \ \le \ C\left(\frac{L}{\sqrt{N}} + \epsilon\right), \tag{14}$$

where $C$ is a constant depending on the function class, $L$ is a Lipschitz-related constant ensuring continuity, $N$ denotes the number of training steps (or effective samples), and $\epsilon$ captures lower-order optimization or approximation error.

**Ignoring $\epsilon$.**  In practice, $\epsilon$ is typically much smaller than the dominant statistical term once optimization has progressed, and is therefore neglected in the following back-of-the-envelope calculation.

**Reference error level.**  When $N = 1$, the bound reduces to

$$\mathbb{E}\big[\|\nabla_a Q_\phi - \nabla_a Q^*\|\big] \ \le \ CL, \tag{15}$$

which corresponds to an untrained or randomly initialized network. We take this quantity as a reference level, corresponding to a 100% relative error under a normalized scale.

**Relative error reduction.**  Suppose we aim to reduce the error to a fraction $\delta \in (0, 1)$ of its initial level.

$$\mathbb{E}\big[\|\nabla_a Q_\phi - \nabla_a Q^*\|\big] \ \le \ \delta\, CL. \tag{16}$$

Substituting the bound and canceling the shared constants $C$ and $L$ (which are fixed for the same network and function class) yields

$$\frac{CL}{\sqrt{N}} \ \le \ \delta CL. \tag{17}$$

Solving for $N$ gives

$$N \ \ge \ \frac{1}{\delta^2}. \tag{18}$$

**Examples.**  For a 1% relative error ($\delta = 0.01$), this requires $N \ge 10^4$ training steps. For a 10% relative error ($\delta = 0.1$), this requires $N \ge 10^2$ training steps. The same calculation applies to an error requirement of 0.1%.This illustrates that even modest relative accuracy targets can correspond to large differences in training cost.

**Discussion.**  This calculation depends on the relative error reduction under fixed function-class constants. Since the quantity under consideration is already normalized to lie in $[0, 1]$, a 1% error directly corresponds to a 1% relative deviation under this normalized scale. This is exactly the same notion of relative error used in the preceding discussion. The specific values of $C$ and $L$ are not critical, as they cancel when comparing different accuracy levels for the same network. The purpose of this appendix is solely to provide an order-of-magnitude intuition for computational savings, rather than a task- or environment-specific error interpretation.

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
