# OpenReview forum: "Modular Diffusion Policy Training: Decoupling and Recombining Guidance and Diffusion for Offline RL"
_TMLR — Rejected by TMLR_

### Review · Reviewer_pyPP · 2025-11-20

**Summary Of Contributions:**

This paper proposes Guidance-First Diffusion Training (GFDT) for diffusion-based offline RL, where the core idea is to decouple the guidance module from the diffusion policy. Specifically, this paper shows that guidance and diffusion modules trained under different algorithms can be interchanged at inference time. Theoretically, intuitions are provided to justify why pretraining guidance on the offline dataset yields an approximately optimal in-distribution value estimator. Empirically, evaluations on D4RL benchmarks are provided to justify the effectiveness of proposed method.

**Additional Comments:**

In addition, the reviewer thinks that the quality of the manuscript can be further improved by including a paragraph to discuss the relation between the proposed methodology and the inference-time scaling/alignment framework for diffusion models [1,2,3,4,5], which turns out to be a popular and highly related topic.

Overall, the reviewer thinks that this paper studied an important topic, but it needs nontrivial amount of refinement before being considered for TMLR. The authors are highly encouraged to take all suggestions above into consideration, provide a detailed discussion and cite missing references listed below.

References:

[1] Uehara, Masatoshi, Yulai Zhao, Chenyu Wang, Xiner Li, Aviv Regev, Sergey Levine, and Tommaso Biancalani. "Inference-time alignment in diffusion models with reward-guided generation: Tutorial and review." arXiv preprint arXiv:2501.09685 (2025).

[2] Singhal, Raghav, Zachary Horvitz, Ryan Teehan, Mengye Ren, Zhou Yu, Kathleen McKeown, and Rajesh Ranganath. "A general framework for inference-time scaling and steering of diffusion models." arXiv preprint arXiv:2501.06848 (2025).

[3] Chen, Haoxuan, Yinuo Ren, Martin Renqiang Min, Lexing Ying, and Zachary Izzo. "Solving inverse problems via diffusion-based priors: An approximation-free ensemble sampling approach." arXiv preprint arXiv:2506.03979 (2025).

[4] Skreta, Marta, Tara Akhound-Sadegh, Viktor Ohanesian, Roberto Bondesan, Alán Aspuru-Guzik, Arnaud Doucet, Rob Brekelmans, Alexander Tong, and Kirill Neklyudov. "Feynman-kac correctors in diffusion: Annealing, guidance, and product of experts." arXiv preprint arXiv:2503.02819 (2025).

[5] Ma, Nanye, Shangyuan Tong, Haolin Jia, Hexiang Hu, Yu-Chuan Su, Mingda Zhang, Xuan Yang et al. "Scaling Inference Time Compute for Diffusion Models." In Proceedings of the Computer Vision and Pattern Recognition Conference, pp. 2523-2534. 2025.

**Audience:**

Yes

**Audience Explanation:**

The paper studies a topic at the intersection of offline RL and diffusion-based policies, which will be of interest to the TMLR community from the reviewer's perspective.

**Claims And Evidence:**

Yes

**Claims Explanation:**

The reviewer thinks that only a subset of the results presented in the paper is supported by clear evidence. Specifically, the theoretical parts are mainly arguments based on intuitions, where there is no rigorous proof. In particular, since Theorem 2 doesn't involve any mathematical terms or rigorous claims, it probably shouldn't be listed as a "theorem" - consider using replacements like "proposition" or "heuristic arguments". Overall, the authors probably should revise the theoretical parts by making it sufficiently rigorous.

**Requested Changes:**

It seems to the reviewer that writing of the manuscript needs to be improved from many aspects. For instance, there are many typos and grammatical issues like “Aross Experiments”, “THe abbrevations”, etc. The authors probably need to read the whole manuscript carefully and make changes if possible. Also, there are many long sentences that are difficult to parse in the paper. The authors are encouraged to read the paper and make revisions if possible. Also, Some figures like the multi-panel plots on pages 22–26 use very small fonts and crowded legends. Probably some simplification is needed here.

---

> ### Author Response · Authors · 2026-01-12
>
> Comment:
> The reviewer thinks that only a subset of the results presented in the paper is supported by clear evidence. Specifically, the theoretical parts are mainly arguments based on intuitions, where there is no rigorous proof.
>
> In particular, since Theorem 2 doesn't involve any mathematical terms or rigorous claims, it probably shouldn't be listed as a theorem''. The reviewer suggests using replacements likeproposition'' or ``heuristic arguments''. Overall, the authors should revise the theoretical parts by making them sufficiently rigorous.
>
> Response:
> We agree with the reviewer that Theorem 2 is not a formal mathematical theorem. In the revised manuscript, we have renamed it as an informal Proposition and clarified its explanatory purpose.
>
> Comment:
> Some figures, such as the multi-panel plots on pages 22–26, use very small fonts and crowded legends. Simplification is recommended.
>
> Response:
> Thank you for the suggestion. We have simplified these figures by enlarging fonts, decluttering legends, and reducing the number of panels where possible to improve readability.
>
> Comment:
> The reviewer suggests adding a paragraph discussing the relationship between the proposed methodology and inference-time scaling/alignment frameworks for diffusion models. Overall, the reviewer believes that this paper studies an important topic, but requires a nontrivial amount of refinement before being considered for TMLR. The authors are encouraged to address all suggestions and cite the missing references.
>
> (We carefully read every paper you mentioned.)
>
> Response:Relation to Inference-Time Alignment. Inference-time alignment for diffusion models as
> policies, enabling controllable generation without retraining the generator [Uehara et al., 2025]. They
> are typically formalized using ordinary, partial, or stochastic differential equations describing diffusion
> sampling dynamics [Dhariwal and Nichol, 2021]. Model-agnostic methods exploit diffusion stochastic-
> ity through restarts, ensemble aggregation, or noise search as inference-time compute scaling [Li et al.,
> 2023, Ma et al., 2025]. Model-specific approaches formulate alignment as sampling from reward-tilted dis-
> tributions modifying the original diffusion objective [Uehara et al., 2025]. Among Model-specific methods,
> Gradient-based methods inject reward, value, or classifier gradients into denoising dynamics, including clas-
> sifier and classifier-free guidance [Dhariwal and Nichol, 2021, Chung et al., 2023b]. Such methods require
> strong alignment between guidance models and diffusion policies, and remain underexplored in offline re-
> inforcement learning [Yang et al., 2023]. Our work contributes is closest to this direction, but also has
> fundamental difference from it. Derivative-free approaches instead post-process generated samples using
> candidate selection, importance sampling, or sequential Monte Carlo resampling [Ramesh et al., 2022b,
> Zhang et al., 2024]. Bayesian SMC methods rely on Feynman–Kac formulations, emphasizing posterior
> correction through weighting rather than stable policy learning [Skreta et al., 2025, Doucet et al., 2009].
> Our work is orthogonal to the research on inference-time alignment, which focuses on goal-oriented
> post-processing, such as selection or resampling, applied to a fixed, pre-trained model. Instead, we modify
> the training and composition of the model itself: we reorder the training pipeline via Guidance-First Dif-
> fusion Training (GFDT) to ensure guidance convergence. We also demonstrate that independently trained
> guidance and diffusion modules can be recombined at inference time without performance degradation.
> This modular training perspective reveals a form of compositional flexibility that is not captured by exist-
> ing inference-time alignment frameworks, and it provides a practical pathway for reusing and recombining
> reinforcement learning policy components across algorithms.
>
> All writing-related improvements are summarized in the response to the reviewer ShzR.

---

### Review · Reviewer_ShzR · 2025-12-09

**Summary Of Contributions:**

The authors proposes Guidance-First Diffusion Training (GFDT).

- Instead of training a diffusion model first and then use it to generate samples for the guidance model, the authors take the opposite: pre-train a guidance module and use the fixed pre-trained module to guide the diffusion process.

- The authors show the framework is modularized and these modules can work in a plug and play fashion.

- The framework is theoretically sound.

- Experimental evaluation is extensive. The ablation study is particularly clear and insightful.

**Audience:**

Yes

**Audience Explanation:**

The RL community may be interested in this.

**Broader Impact Concerns:**

Not needed.

**Claims And Evidence:**

Yes

**Claims Explanation:**

The paper’s key claims are well supported by experimental evidence. For example, Table 2 demonstrates that GFDT consistently outperforms existing methods on nearly all benchmark datasets.

The ablation study in Section 4.1.1 further validates the framework’s design choices by isolating the role of the guidance module. In particular, unfreezing the guidance during diffusion training leads to significantly higher variance and worse performance, directly confirming the two challenges outlined in the introduction. Similar degradations are observed when the guidance module is removed or replaced.

Section 4.2 provides empirical evidence for the plug-and-play property, one of the paper’s central contributions. The results show that independently trained guidance and diffusion modules can be recombined without joint training while still achieving competitive performance.

**Requested Changes:**

I think the paper is in a good shape overall. One area that could be improved is the clarity of the figures. For instance, Figures 1 and 3 contain complex flows of data and control signals, and the current visual design makes them somewhat difficult to parse. Providing more detailed captions or annotating the diagrams more explicitly would help readers understand the modular interactions and training/inference pathways more easily.

Also, in Table 2, variants like EDP_GFDT, DGID and IGDD show very large variance in performance across datasets. Can the authors provide some clarifications?

---

> ### Author Response · Authors · 2026-01-12
>
> Comment:
>  One area that could be improved is the clarity of the figures. Figures~1 and~3 contain complex flows of data and control signals, and the current visual design makes them difficult to parse.
>
> We agree that the modular interactions and the training/inference pathways in the Methodology diagrams could benefit from more explicit annotations.
> In the revised manuscript, we have updated the diagrams by adding clearer annotations and expanding the figure captions to explicitly distinguish the roles of the diffusion model and the guidance module during training and inference.
> In addition, we introduced a dedicated Notation and Terminology table that explicitly defines all variables and modules used in the Methodology, and we added references in the main text directing readers to this table for clarification.
>
> We have revised Figure 1 to improve clarity by simplifying the visualization and removing implementation-level details (e.g., plug-and-play tuning components), while retaining only the core mechanism of exchanging IDQL and DQL guidance.
> Moreover, instead of presenting a single end-to-end GFDT pipeline, the revised figure explicitly contrasts different components of GFDT with the conventional classifier-free guidance framework, which we believe makes the conceptual differences clearer.
> We also added accompanying pseudocode for further clarification; due to rebuttal system limitations, the updated figure cannot be shown here, but will be included in the revised manuscript.
> Comment:
> Providing more detailed captions or more explicit annotations in the diagrams would help readers better understand the modular interactions and training/inference pathways.
>
> Response:
> Thank you for the suggestion. We agree that the Methodology section involves a relatively large number of variables and modular components, which may make the training and inference pathways harder to follow.
> To address this, we have revised the manuscript in two ways.
> First, we expanded the captions and annotations in the Methodology diagrams to explicitly clarify the roles of each module and their interactions during training and inference.
> Second, we added a dedicated Terminology and Notation that explicitly defines every variable and symbol used in the Methodology, including the diffusion model, guidance module, pretrained components, and their associated parameters.
>
> We thank the reviewer for pointing out the relatively large variance observed for several variants (e.g., EDP_GFDT, DGID, and IGDD).
> This variance is mainly driven by random instability in certain continuous-control environments.
> In environments such as HalfCheetah and Hopper, performance is highly sensitive to early-stage behavior.
> For some initial conditions, the agent may terminate almost immediately (e.g., falling at the beginning of an episode), leading to near-zero returns.
> Even a small number of such early failures can substantially increase the reported variance, despite most runs achieving stable and competitive performance.
>
> In addition, EDP relies on one-step denoising, which has higher sensitivity on the guidance signal and makes performance more sensitive to small changes or inaccuracies in guidance. Overall, the observed variance reflects the inherent instability of these benchmarks and sensitivity to rare early failures, rather than inconsistent behavior across runs.
>
> Reviewer Comment: Request for writing quality improvement.
>  Response:
> We sincerely thank the reviewer for highlighting these important shortcomings. In the revised version, we have undertaken a comprehensive rewrite to address each point raised:
>
> Language and grammar – We thoroughly proofread the manuscript to correct typos and grammatical errors, and simplified complex or lengthy sentences throughout.
>
> Structure and readability – We reorganized the paper to improve flow and clarity. Specifically, we moved the full pseudocode and detailed description of GFDT into the introduction so that the method is clearly defined before comparisons are made.
>
> Accessibility for external readers – We integrated key background material from the appendices into the main text (including essential explanations of Q-guided diffusion and diffusion policy frameworks), while keeping technical derivations in the appendix for completeness. This ensures the core ideas are now self-contained in the main paper.
>
> Theoretical presentation – We downgraded the overly strong “Theorem 2” to an informal proposition, clarified its illustrative purpose under idealized assumptions, and explicitly discussed its limitations in practical settings (e.g., deterministic MDPs, coherent batches). We also added citations and explanations for Equation (5) to clarify its role as an intuitive tool rather than a formal bound.
>
> Methodological clarity – We added a new figure (Figure 1) in the introduction to visually contrast GFDT with traditional classifier-free guidance, making the contribution and mechanism immediately clear to readers.

---

### Review · Reviewer_fzin · 2026-01-02

**Summary Of Contributions:**

This work proposes a merger of Q-guided diffusion and diffusion policy frameworks.

The experimental scope is broad and clearly exposes the method's advantages and limitations.

The paper provides a clear discussion of the method's applicable scope and limits.

**Audience:**

Yes

**Audience Explanation:**

RL and diffusion models are entirely in the scope of TMLR.

**Claims And Evidence:**

No

**Claims Explanation:**

- I think the algorithmic novelty seems moderate - while I also emphasize that I am not at all a subject expert here. The method seems to build on existing Q-guided diffusion and diffusion policy frameworks, and both the diffusion component and the Q-learning component follow known designs. Feel free to correct me if I am wrong.

- There are some notational issues that can confuse the role of the critic versus the diffusion model, and the theoretical assumptions do not strictly hold in this problem setting, but this is not fully discussed.

- Some key theorems are stated without citation or proof sketch, so their conditions of validity are unclear.

- In addition, Figure~1 shows undertrained / partially trained / fully trained guidance and diffusion, but these regimes are not formally defined in the text. The paper does not clearly explain under which conditions a guidance network should be regarded as undertrained or how exactly such an undertrained guidance is expected to affect diffusion, beyond a brief high-level statement on page 4 that before sufficient training an inaccurate $Q$ can add incorrect gradients.

**Requested Changes:**

0. First and foremost, the paper needs to written to at least somewhat help ``external'' audiences like me. The background should ideally include a sufficient ground-up introduction to the key background ideas of Q-guided diffusion and diffusion policy frameworks, on which this work seems to build on. I can see some reviews in Appendices A and C but it seems much of that should be part of the main paper to help the external readers.

    Also the pseudocode for GFDT and its detailed description should be a part of Section 1, otherwise we currently have a strange format whereby GFDT is being compared other things in Section 2, but by then as a reader I have never been given a detailed description of what that is.

1. In Equation (2), the paper writes $\nabla_\theta Q(s,a)$. Since $\theta$ denotes diffusion parameters (guessing so) and $Q$ is parameterized by $\phi$, I think this should be $\nabla_{\phi} Q_{\phi}(s,a)$. Also have the $Q$ and $L_{total}$ and $L_{BC}$ notation used in equation 2 been formally defined anywhere where before this? I think not.

2. The theorem seems to rely on deterministic MDPs, coherent batches, and a particular batch-constrained Bellman operator. Are these conditions met on the D4RL test with function approximation? The paper already notes that the assumptions are not fully satisfied; it would be better to discuss how to interpret the theorem in this setting.

   And where is the proof of Theorem 2? Is this even a mathematically rigorous theorem? (It uses a quantity called $Q_\phi$ which doesnt seem to have been precisely defined.)If this is a critical thing then it should be a separate section on theoretical guarantees.

3. Equation (5) has no citation or proof. Please add a precise source or an appendix argument under the paper's setting, including the dependence on the Lipschitz constant of $Q_{\phi}$, the diffusion noise scale, and the step-size bound.

4. The directional-alignment assumption is strong in offline RL with function approximation. Please discuss when alignment can be negative and how that would affect the improvement claim.

5. To match the guidance-first claim, please add a minimal ablation showing how an undertrained guidance affects training at different stages, and how a converged guidance helps.

6. It would be helpful to add clear definitions of undertrained / partially trained / fully trained guidance (and diffuser). As an external reader I am quite inclined to read the diagram in details to be able to understand the paper.

    Additionally, there should be a more precise explanation or derivation of why, under your theoretical conditions, an undertrained guidance network is expected to misguide diffusion training, assuming I am understanding this correctly.

---

> ### Author Response · Authors · 2026-01-12
> **The main comments concern (1) the novelty of the proposed method, (2) the notation and role of the Q-based guidance module, (3) the interpretation of Figure 1, and (4) the relationship between our terminology and the actor–critic framework. We address these points in detail below.**
>
> Thanks for the comment.
>
> Comment:  “The algorithmic novelty seems moderate ...”
>
> Response: We agree that our contribution is not a new loss function or architecture.
> Instead, the novelty is methodological and mechanism explanation. Our work proposed guidance-first modular training in offline RL. This idea is intuitive but has not been applied in prior offline RL studies.
> Directly applying an arbitrary high-performance guidance to diffusion training can lead to empirical instability as shown in Section 5, which may be a reason why GFDT is not applied in existing research. Our research is the first to clarify the improvements and limitations of guidance-first training. Existing research in the image area has also explored training methods using an external guidance module, such as classifier guided generation. However, there is no clarification of application scope of guided training in image region as well. Therefore, this paper's clarification of application scope is impactful both within and outside the realm of offline RL. As shown in our Section 5, the key issue is distribution mismatch. Guidance and diffusion must be trained on the same data distribution. When the guidance is out-of-distribution, offline RL training fails. Under distributional alignment, frozen guidance can reliably guide diffusion policies. Our research proposed a method that reduces training cost and improves performance. Importantly, it defines a clear applicability boundary.
>
> Similar to how blood types enable safe transfusion, we identify when modular guidance is beneficial to use, a principled condition for the components can be composed reliably.
>
> comment: In Equation (2), the paper writes $\nabla_\theta Q(s, a)$. ..I think not.
>
> Response: Thank you very much for the careful reading and for pointing out the potential ambiguity in our notation. We apologize that this part was not stated clearly enough in the original version. We agree that we should have explicitly clarified the roles of the symbols at this point in the paper, as well as summarized them more clearly in a terminology section, which we have now added in the revised version.
>
> To clarify, both $Q_\theta$ and $Q_\phi$ denote Q-networks used as value-based guidance modules, whose role is to provide reward-aware guidance to the diffusion process. The diffusion component itself is consistently represented as $f_\textdiffusion$, which serves as the generative module and is not a Q-based network. The use of the leading symbol $Q$ is intentional and indicates that these networks estimate value functions rather than perform diffusion or generation.
>
> The distinction between $Q_\theta$ and $Q_\phi$ is purposeful. $Q_\theta$ refers to a guidance network that is trained and optimized within the current learning pipeline using the dataset under consideration. In contrast, in Proposition~2 (formerly Theorem~2), we introduce $Q_\phi$ to explicitly represent a pretrained guidance network, which may be obtained externally and directly reused without further optimization. These two symbols therefore do not correspond to different model types (e.g., diffusion versus guidance), but rather to different training statuses and origins of guidance networks.
>
> We acknowledge that this distinction was not sufficiently emphasized before, and we have revised the manuscript to make it explicit both locally around Proposition~2 and globally in the terminology section. We thank the reviewer again for the careful reading, which helped us significantly improve the clarity of the paper.
>
> Comment: Some key theorems are...how exactly such undertrained guidance is expected to affect diffusion ...
>
> Response: We apologize that we did not explicitly state before Figure 1 that it is an illustrative (conceptual) figure, which may have led to it being misinterpreted as an architecture diagram. We have revised the paper to clarify this. Figure 1 is intended to provide intuition on how guidance and diffusion mature over training and when guidance is injected in different methods. This figure is not to define ablation stages. Regarding completely untrained guidance, we include random guidance experiments, which show that random guidance destroys training and yields negative normalized reward, highlighting the necessity of meaningful guidance.
> Comment: There are some notational issues that can confuse the role of the critic versus the diffusion model
>
> Response:
> In literature, diffusion-based policies are sometimes interpreted as actors and Q-based guidance modules as critics; however, we adopt the terms diffusion model and guidance module to emphasize that our method does not rely on standard actor–critic training dynamics, as the guidance module is trained independently and only affects sampling.
> To avoid potential confusion, we have added a clarification in the last paragraph of the introduction.
> Additional writing-clarity comments are addressed under Reviewer ShzR; proof-rigor details under pyPP.

---

### Decision · Action_Editor_U9Jn · 2026-02-21

**Recommendation:** Reject

**Additional Comments:**

Reviewers noted several merits of this submission, including its timely topic and experimental demonstrations.  However, they also pointed to several major shortcomings, including an informal theoretical presentation (inconsistent with claims for rigorous theory), various issues concerning writing, and a possibility for AI-hallucinated citations (e.g., Uehara et al. 2025 and Skreta et al. 2025).  In light of this, it is my opinion that the submission requires a major revision for publication-readiness.

**Audience:**

Yes

**Audience Explanation:**

The topic of the submission (offline RL with diffusion-based policies) is of interest to the community.

**Claims And Evidence:**

No

**Claims Explanation:**

While there is generally a consensus that the submission backs its empirical claims, the papers makes theoretical claims that do not seem to be properly supported (e.g., it claimed that results are proven but the reviewers were not able to trace formal proofs).

**Resubmission Of Major Revision:**

The authors may consider submitting a major revision at a later time.